# Learning causal networks using inducible transcription factors and transcriptome-wide time series

Sean R Hackett[1] (iD), Edward A Baltz[2] (iD), Marc Coram[2] (iD), Bernd J Wranik[1], Griffin Kim[1], Adam Baker[1], Minjie Fan[2], David G Hendrickson[1], Marc Berndl[2] & R Scott McIsaac[1,*] (iD)

## Abstract

We present IDEA (the Induction Dynamics gene Expression Atlas), a dataset constructed by independently inducing hundreds of transcription factors (TFs) and measuring timecourses of the resulting gene expression responses in budding yeast. Each experiment captures a regulatory cascade connecting a single induced regulator to the genes it causally regulates. We discuss the regulatory cascade of a single TF, Aft1, in detail; however, IDEA contains > 200 TF induction experiments with 20 million individual observations and 100,000 signal-containing dynamic responses. As an application of IDEA, we integrate all timecourses into a whole-cell transcriptional model, which is used to predict and validate multiple new and underappreciated transcriptional regulators. We also find that the magnitudes of coefficients in this model are predictive of genetic interaction profile similarities. In addition to being a resource for exploring regulatory connectivity between TFs and their target genes, our modeling approach shows that combining rapid perturbations of individual genes with genome-scale time-series measurements is an effective strategy for elucidating gene regulatory networks.

**Keywords** causal inference; gene expression; transcription factors; whole-cell modeling; dynamical systems

**Subject Categories** Chromatin, Transcription & Genomics; Computational Biology; Methods & Resources

**Mol Syst Biol. (2020) 16: e9174**

## Introduction

A central problem in modern genomics is how to extract causality from experimental data (Maathuis *et al*, 2010; Davey Smith & Hemani, 2014). When a cause–effect relationship can be established, direct effects usually cannot easily be discriminated from indirect effects, thereby limiting interpretability. Causality establishes a regulator's potential to alter the level or activity of a downstream target; directness increases the likelihood of information being transmitted without adulteration.

The direct and indirect molecular interactions that achieve a particular cellular state can be described as regulatory edges that collectively form gene regulatory networks (GRNs) (Shen-Orr *et al*, 2002; Alon, 2006). As genome-scale datasets started to become available over 20 years ago, work by Alon and colleagues established that certain GRN topologies are enriched in biological systems (Shen-Orr *et al*, 2002). Understanding the functional properties of such "network motifs" became the subject of intense experimental and theoretical investigation (Milo, 2002; Shen-Orr *et al*, 2002; Mangan *et al*, 2003, 2006; Eichenberger *et al*, 2004; Milo *et al*, 2004; Alon, 2007; Goentoro & Kirschner, 2009; Goentoro *et al*, 2009; Shoval *et al*, 2010; Murugan, 2012). Combined with genomic tools and extensive prior knowledge, it became possible to identify network motifs/GRNs associated with core cellular processes, with early work in yeast focusing on cell cycle control and the DNA damage response (Simon *et al*, 2001; Workman *et al*, 2006). The widespread development and adoption of genome-scale technologies, including the creation of mutant libraries and the power of CRISPR-Cas systems, have further enabled GRN discovery across organisms, from plants (Chen *et al*, 2018), to yeast (Kemmeren *et al*, 2014), to humans (Rubin *et al*, 2019).

How are genomic approaches commonly applied to identify GRNs, and what are their limitations? One approach is to measure gene expression profiles of deletion mutants (Kemmeren *et al*, 2014). Without dynamics, however, there is limited potential for determining *direct* regulatory relationships because information propagation from the deletion to each differentially expressed gene is not observed (Kang *et al*, 2020). The final measurement (the gene expression profile) is the asymptotic readout of many layers of regulation and unobserved (but potentially relevant) molecular interactions. Using methods like ChIP-seq or ChIP-exo provides another strategy for determining GRNs that focus on transcription factors (TFs) and their target genes (Harbison *et al*, 2004; Gerstein *et al*, 2010; Nègre *et al*,

1 Calico Life Sciences LLC, South San Francisco, CA, USA
2 Google Research, Mountain View, CA, USA
 *Corresponding author. Tel: +1 650 769 5539; E-mail: rsm@calicolabs.com

2011; Kheradpour & Kellis, 2014; Kim *et al*, 2014; Kang *et al*, 2020). Target genes with similar ChIP profiles can exhibit opposite expression responses (Lickwar *et al*, 2012), and highly expressed portions of the genome can exhibit strong ChIP signal even among unrelated proteins (Teytelman *et al*, 2013). Interpreting the biological importance of such peaks must be done with sufficient controls to distinguish whether signals are truly biological vs. technical in origin, but the challenge remains that ChIP-based approaches alone provide no assessment of TF functionality. Integration of multiple 'omic technologies combined with time-series measurements can help identify direct functional interactions to elucidate GRNs, as was done in a recent study that combined RNA-seq, NET-seq, and ChIP-seq to identify a core regulon for Hsf1 in yeast (Solís *et al*, 2016). Finally, there is a growing literature of computational methods for reconstructing GRNs from high-throughput data (Pe'er *et al*, 2001; Markowetz *et al*, 2007; Stolovitzky *et al*, 2009; Marbach *et al*, 2010, 2012a,b; Yip *et al*, 2010; Yang *et al*, 2018). The Dialogue on Reverse Engineering Assessment and Methods (DREAM) project, which is organized around annual challenges, provides a framework to benchmark network inference methods (Marbach *et al*, 2010). Network inference performance can depend on implementation as well as the network structure itself (Marbach *et al*, 2012a). In the DREAM5 challenge, no single inference method performed optimally across multiple datasets. Integrating predictions across all participating teams (35 inference methods in total) to generate "community networks" had the most robust performance (Marbach *et al*, 2012a).

We propose an alternative approach to revealing GRNs. Within a complex network, we believe that identifying new, direct, *and* functional regulatory relationships is best facilitated by activating one network element followed by dynamic measurement of all other network elements. A seminal paper from Chua *et al* (2006) revealed that overexpression of TFs, followed by transcriptome profiling at a single time point, can reveal functional regulator-gene connections that are absent when profiling TF-deletion mutants. Following that work, we combined TF activation with dynamic transcriptome profiling to dissect the incompletely understood regulatory connectivity of the yeast sulfur regulon (McIsaac *et al*, 2012). By generating strains that separately expressed each known sulfur-related TF (Met4, Met28, Met31, Met32, or Cbf1) from an engineered promoter that is strongly repressed, but then activated by a small molecule (β-estradiol), a single TF was rapidly induced and genome-wide transcriptional responses were tracked over time. This approach enabled the identification of a novel feed-forward loop between Met4 and Met32, dozens of new instances of feedback, clear subfunctionalization of Met31 and Met32 (paralogous TFs that bind to the same sequence) (McIsaac *et al*, 2012), and revealed that Cbf1 could act as an activator or a repressor, depending on which promoter it targets (McIsaac *et al*, 2012). For certain methionine metabolic genes, Cbf1 can act as an activator of target genes when yeast are limited for methionine, but can switch to being a repressor of those same genes when yeast are limited for phosphate and have excess extracellular methionine (McIsaac *et al*, 2012). Thus, TF induction followed by dynamic transcriptome profiling can reveal condition-dependent regulatory connections, and a single TF can act as both a positive and a negative regulator of gene expression depending on local DNA context and environmental conditions.

More recently, this activation-based approach was used to determine that a particular single-nucleotide polymorphism was the true causal variant underlying an expression quantitative trait locus (eQTL) (Lutz *et al*, 2019). Genome-wide association studies can reveal regions of the genome of interest that are important for a phenotype; targeted perturbations of individual genes followed by dynamic expression profiling can determine whether a genetic variant is causal. Despite the successes of these studies, genome-wide time-resolved datasets following gene induction remain uncommon. New experimental datasets and analytical approaches are required for revealing GRNs and for learning non-canonical regulators at the scale of the entire genome.

We describe the creation of IDEA (the Induction Dynamics gene Expression Atlas) and demonstrate the value of this dataset for revealing both new and known genome-scale causal relationships. We generated ~ 200 induction experiments in which a single yeast TF was rapidly induced from a β-estradiol-responsive synthetic promoter, and full transcriptome differential expression was subsequently tracked, typically across eight time points. Such experiments feature the near immediate strong induction of an inducer-driven TF of interest, followed by rapid changes in genes that are directly regulated by these TFs, and later changes of indirectly regulated genes (Fig 1A–C). While these indirect effects contain many uncharacterized regulatory processes, they can be difficult to attribute to a specific regulator (Fig 1D) using a single time-resolved experiment. By aggregating all experiments (each of which includes kinetic information), we can potentially determine which regulator(s) are acting in individual timecourses by identifying the parsimonious set of regulators whose abundances account for each gene's expression variability (Fig 1E and F). Our approach implicitly dissects indirect regulation into a series of direct regulatory relationships. Predicted intermediate regulators span canonical transcriptional regulators and genes of unknown function. We successfully validated three underappreciated regulatory hubs among 10 model-predicted latent regulators.

## Results

Each of 201 genes' native promoters was separately replaced with a β-estradiol-inducible promoter as previously described (McIsaac *et al*, 2012, 2013; Table EV1). This set of induced genes is heavily enriched for non-essential TFs and chromatin modifiers. Each strain was grown to a steady state in chemostat culture and following the addition of β-estradiol, the full transcriptome was measured at 4–10 post-induction time points (83% of experiments contain eight time points). We chose chemostats, in part, because the steady-state condition of chemostat cultures is a particularly useful feature for mathematical modeling. Under steady-state conditions, the levels of molecules and activities of processes are not changing at a culture-wide level. Therefore, following TF induction in a steady-state culture, immediate dynamic changes result from the TF induction itself. The ability to choose a single growth-limiting nutrient also makes the chemostat ideal for exploring how input–output relationships between TFs and target genes vary under different nutritional conditions (McIsaac *et al*, 2012).

In total, IDEA contains > 1,650 microarrays comprising 217 distinct induction experiments. A subset of these data (six TFs) that focused on the sulfur regulon and amino acid metabolism (Cbf1, Gcn4, Met4 Met28, Met31, and Met32) was published previously (McIsaac *et al*,

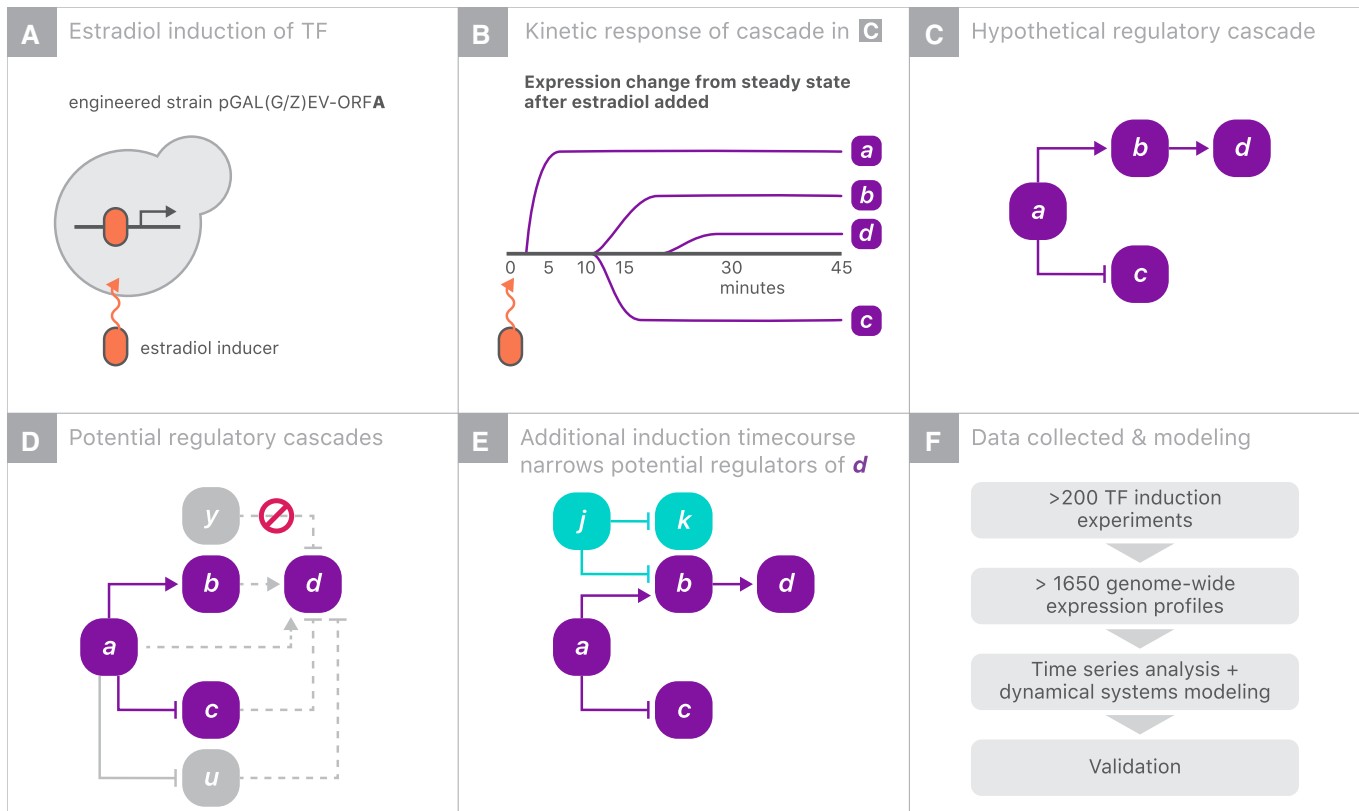

**Figure 1.  Inferring direct regulation using many TF induction experiments.**

A   In each experiment, one transcriptional regulator with an inducible promoter is rapidly overexpressed in response to 1 μM β-estradiol.

B   Example of three genes (labeled B, C, and D) responding with different kinetics following induction of regulator A.

C   Hypothetical example of a regulatory cascade in which an induced transcriptional regulator A directly inhibits C and directly activates B. B, in turn, directly activates D.

D   In practice, we do not know that A regulates D via B and instead want to infer such regulatory relationships. In this example, direct regulation of D by B is only one hypothesis that is consistent with the data—all viable hypotheses are shown by dashed lines. A could directly activate D, C could inhibit D, or A could regulate an unmeasured confounder U which is the true regulator. Direct regulation by a variable Y, which is independent of A, is not possible since the timecourse begins at steady state.

E   Integrating the A induction timecourse with a second induction timecourse, which perturbs B without perturbing A or C, allows us to narrow down D's possible sources of regulation. In this case, U may still be a possibility if it is remains correlated with B.

F   Overview of data and analysis performed in this study. Over 200 induction experiments were constructed allowing for many opportunities to resolve ambiguous regulation.

2011, 2012, 2013). Fifteen induction experiments were repeated at least once using the same induced gene to either capture late changes in some experiments, confirm similar expression response to two different induction systems [referred to as ZEV (McIsaac *et al*, 2013, 2014) and GEV (McIsaac *et al*, 2011)], or to verify reproducibility (Appendix Fig S1A). The majority of experiments were sampled at $t = 0, 5, 10, 15, 20, 30, 45$, and 90 min following β-estradiol addition in phosphate-limited chemostats. Induction of a target TF is detectable in < 5 min and reaches saturation within ∼ 10 min following β-estradiol addition at a median level 53-fold higher than at $t = 0$ min (Appendix Fig S1B), which, in terms of magnitude, is less than the ∼500- to 1,000-fold inducibility of Gal4-driven promoters in the presence of galactose and similar to the ∼50- to 100-fold repressibility of nitrogen catabolite repression (NCR) genes in response to an increase in nitrogen availability (Johnston, 1987; Biggar & Crabtree, 2001; Airoldi *et al*, 2016). In IDEA, we estimate that 86% of synthetic promoter-driven TF alleles have lower expression than native

promoter-driven TF alleles (Appendix Fig S2). We estimate that at 90 min following β-estradiol addition, the median TF is 28.4-fold more highly expressed than the native promoter-driven TF allele (Appendix Fig S2). Finally, IDEA also contains several TFs induced under multiple conditions. Gln3, Dal80, and Gzf3 were induced under phosphate and nitrogen limitation (with ammonium sulfate used as the sole nitrogen source in both cases). For each TF, the resulting expression patterns are strikingly similar in the two tested environments (Appendix Fig S3), suggesting that the activity of nitrogen-related TFs may depend more on the quality of the nitrogen source (proline vs. ammonium sulfate, for example), rather than the choice of growth-limiting nutrient. In this manuscript, an *experiment* refers to all of the gene expression responses that follow from induction of single TF. A *timecourse* refers to the kinetic response of a single gene within a single experiment.

Across our dataset, most genes' expression does not change in a typical induction experiment, with some notable exceptions (such

as *FMP48*). Accordingly, the inducer-driven signal of interest is relatively sparse and interspersed among ubiquitous noise. This noise is governed by both a mild stress response (Gasch *et al*, 2000) and log-normal noise that varies across both genes and arrays. In order to isolate inducer-specific expression changes, the stress response was subtracted from each experiment and then an observation-level noise model was used to select a subset of timecourses that are statistically inconsistent with experimental noise (Appendix Figs S4–S6). The signals from these 100,036 timecourses were retained (~ 8% of timecourses), while all other timecourses were set as invariant (a $\log_2$ fold change of zero). The full transcriptional dataset is available as Dataset EV1. Further details on processing these data can be found in the Appendix.

### Transcriptional responses vary in amplitude, kinetics, and shape

The Aft1 experiment is an illustrative example of the value of induction data for revealing intricate regulatory phenomena. Aft1 was originally identified as an activator of genes that uptake iron into the cell (Yamaguchi-Iwai *et al*, 1995). Aft1 responds to defects in iron–sulfur cluster biogenesis (Chen *et al*, 2004), and its activity is negatively regulated by Met4, the primary activator of methionine biosynthetic genes (Chen *et al*, 2004; Petti *et al*, 2012). We highlight Aft1, in part, because we observe a range of expression responses following its activation. Specifically, when Aft1 is induced, two broad classes of expression changes are observed: Fast induction of targets reported to be bound by Aft1 based on ChIP and gradual changes of genes whose expression has previously been shown to be correlated with, but not bound by Aft1 (Fig 2A; Teixeira *et al*, 2018). Additionally, certain timecourses appear sigmoidal, while others possess more complicated dynamics (Fig 2A).

As is the case in the Aft1 experiment, timecourses with significant signal across IDEA typically exhibit either a sigmoidal or a impulse-like response (double sigmoidal); thus, we fit a Bayesian version of the Chechik & Koller (CK) kinetic model to each timecourse (Chechik *et al*, 2008; Chechik & Koller, 2009; see Materials

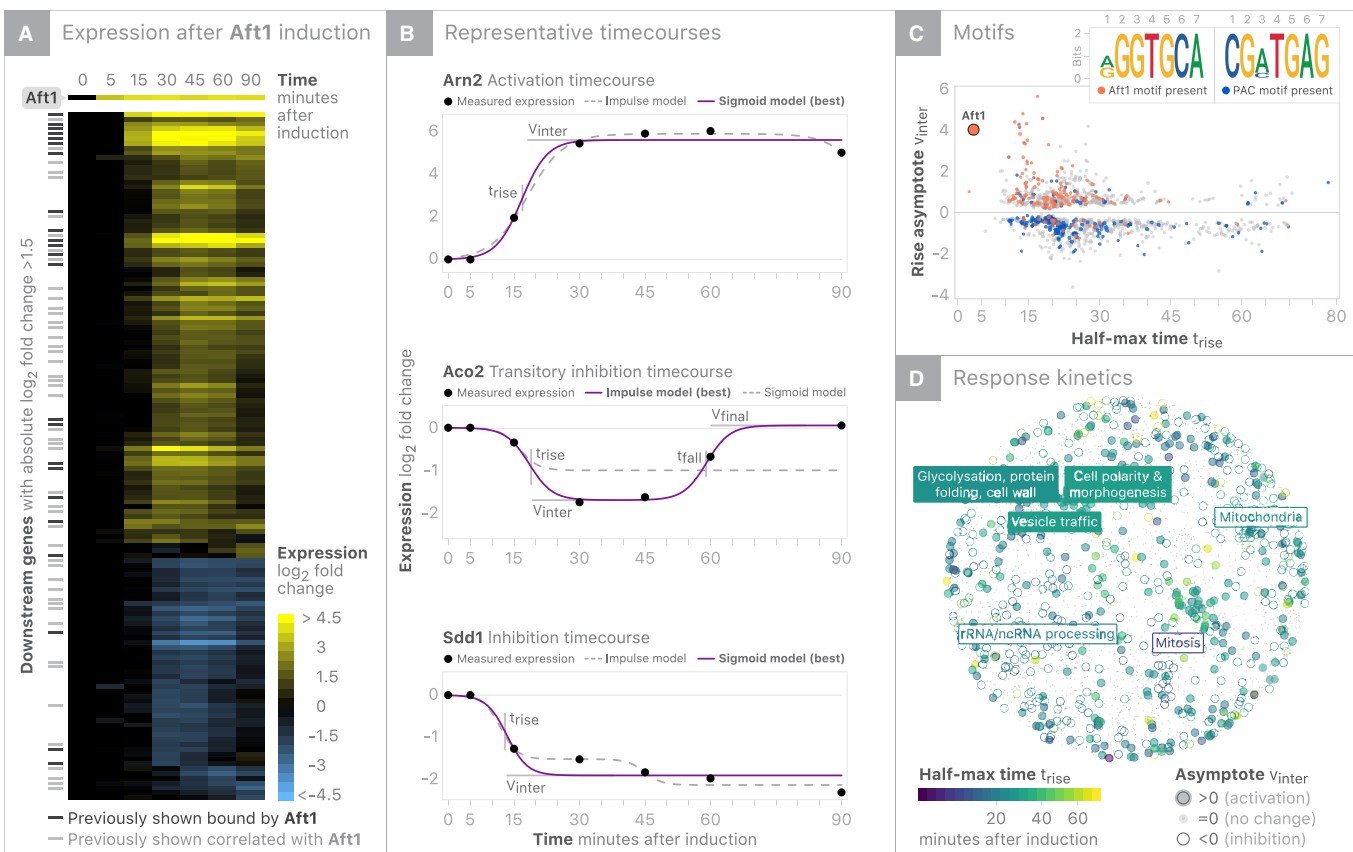

**Figure 2. Characterizing the downstream responses of Aft1 induction.**

A   Heatmap summary of Aft1 induction experiment showing all genes that change with $|\log_2(\text{fold change})| > 1.5$.

B   Parametric summaries of representative sigmoidal activation and inhibition timecourses and impulse (double sigmoid) modeling of transitory inhibition. Sigmoids are summarized by half-max time ($t_{rise}$) and the asymptote ($v_{inter}$), while impulses include a second half-max ($t_{fall}$) time and final asymptote ($v_{final}$). The strongest supported model for each timecourse is shown as a filled in line, while the alternative model is shown with a dashed line.

C   K-mers enriched in the promoters of regulated genes are overlaid on summary of each gene's $t_{rise}$ and $v_{inter}$. The presence of the Aft1 motif is associated with early activation, while early inhibition is associated with the PAC motif (Tod6/Dot6).

D   Response kinetics are overlaid on gene coordinates based on genetic interaction as a surrogate for functional similarity. Up-regulated genes are enriched for vesicle trafficking/glycosylation/polarity processes, and down-regulated genes are enriched for mitochondrial/mitotic/rRNA processes.

and Methods for more details on curve fitting; code for implementing CK fits can be found at https://github.com/calico/impulse). The CK model characterizes a timecourse as a double sigmoid but can be reduced to a simpler sigmoid that has fewer parameters. Specifically, the original CK kinetic model contains six parameters, which we reduced to five parameters because the initial amplitude for all timecourses is zero due to normalization. The impulse (double sigmoid) response is ideal for capturing two-transition behavior in biological timecourses. One sigmoid characterizes the onset response, and a second sigmoid characterizes the offset response (Chechik & Koller, 2009). Parametric fits enable direct comparisons of timecourses by revealing kinetic parameters. Our Bayesian implementation ensures that these parameters are interpretable by penalizing unrealistic and impossible parameterizations (e.g., step-function responses or changes which precede β-estradiol introduction). Since the impulse and single sigmoid models are nested (i.e., the simpler model contains all of the terms within the more complex model), we can—for a given timecourse—use a likelihood-ratio test to determine whether extra parameters improve the fit sufficiently to justify the more complex model.

Sigmoidal responses are summarized with a half-max time constant ($t_{rise}$), an asymptotic expression level ($v_{inter}$), and a slope parameter (β). Impulses include two additional parameters: $t_{fall}$, which describes the time when the response returns halfway to its final level, and $v_{final}$, the asymptotic expression level of the impulse (Fig 2B; Chechik *et al*, 2008). Utilizing these kinetic parameters, we observed multiple binding motifs of genes associated with characteristic response kinetics, including, as expected, the Aft1 motif associated with early activation, and a different motif [recognized by Tod6/Dot6 (also referred to as a PAC motif)] associated with early inhibition (Fig 2C). Targets activated and repressed in the Aft1 experiment have similar kinetic responses, and both classes contain examples of impulse-like expression responses (Fig 2D). Beyond Aft1, other TF experiments that contain a large number of impulse-like responses (indicative of feedback control/perfect adaptation) include Pho4, Mac1, Oaf1, Rtg1, Rtg2, Stb5, and Zap1 (Appendix Figs S7–S9). In total, we find evidence of transcriptional feedback in more than 1,700 timecourses (~ 2% of all timecourses). To allow others to provide comparable investigations into the kinetics, functional coherence, and regulation of each timecourse in our dataset, we provide an interactive website (https://idea.research.calicolabs.com).

We can broadly categorize timecourses at a dataset level based on existing knowledge. While strong acute regulation events are frequently associated with the direct binding of the induced TF (Kang *et al*, 2020), over 75% of genes responding in our dataset are new regulatory connections (Appendix Fig S10). Additionally, we find that 79% of genes reported as being directly bound by a TF based on published ChIP measurements do not exhibit a significant expression response in the corresponding TF's induction experiment (Appendix Fig S10; Teixeira *et al*, 2018). The low recall of reported transcriptional regulation underscores the value of dynamic data. Realized regulation may be impacted by chromatin accessibility and the regulatory context of the extracellular environment, which can result in different post-translational modifications of TFs (Song *et al*, 2011; Arvey *et al*, 2012; McIsaac *et al*, 2012; Gomes & Wang, 2016). This is further supported by the weak agreement between the reported binding and coexpression partners of a TF with the number of genes that change when it is induced (Appendix Fig S11).

## Analysis of promoter composition

Induced TFs directly account for only a portion of the observed expression changes in IDEA, leading to an obvious question: *Which regulators are actually acting in each induction experiment?* To investigate whether the kinetics of responding genes can be informed by promoter composition, we carried out systematic *de novo* motif discovery of all experiments and identified 715 promoter motifs enriched in the responding genes across all experiments (Table EV2). 51% of these motifs could be matched to known regulators and thus suggest plausible candidates for regulators which may operate in each experiment (Weirauch *et al*, 2014; Teixeira *et al*, 2018). While linking TFs to their targets using motifs has been a common assumption in order to enable genome-scale GRN inference, we find this assumption can be limiting. Indeed, in the Aft1 induction experiment, the Aft1 motif is associated with direct activation of only a small number of genes. Since we would like to understand regulatory cascades without requiring regulators to possess direct DNA binding ability, we developed a model that, assuming no prior information, could allow for the elucidation of regulators with unappreciated transcriptional impacts.

## Integrative modeling using IDEA

In a TF induction experiment, we would like to infer which early-responding gene(s) are causally responsible for gene expression changes occurring later in the experiment. In a single experiment, however, we would only be able to identify a coexpressed cluster of genes whose expression coincides with a late change, rather than a single candidate regulator (Fig 1D). While reliably inferring regulatory mediators from a single experiment is a dubious prospect, across all experiments, genes respond in a median of 12 induction experiments (or in ~ 5% of experiments; Appendix Fig S12). Therefore, aggregating multiple experiments provides the potential to decouple each gene's expression dynamics from those of spurious correlates. As we have generated hundreds of strong orthogonal gene-level perturbations, our dataset provides an opportunity to test this approach. Across > 1,650 samples, most genes have a distinct pattern of variation, and such distinctness decays quickly once an appreciable fraction of experiments are removed (Appendix Fig S13).

To learn direct regulatory relationships from such data, we formulated a set of gene-level regression models that predict the rate of change of each target gene as a sparse linear combination of all genes' expression:

$$\Delta \ln(y_{ijt})/\Delta t = \sum_k \left( \alpha_{ik}(y_{kjt} - 1) + \beta_{ik}(y_{ijt}y_{kjt} - 1) \right)/y_{ijt} \qquad (1)$$

Here, $y_{ijt}$ is the expression relative to the control strain at time zero of a gene $i$ in experiment $j$ at a time $t$ (i.e., for treatment $r$ and control $g$, $y_{ijt} = (r_{ijt}/g_{ijt})/(r_{ij0}/g_{ij0})$; therefore, $y_{ij0} = 1\ \forall\{I,J\}$). Here, α represents the linear effect of one transcript on another (i.e., the $k$'th transcript on the $i$'th transcript) and β represents the effect proportional to the target transcript. We allow any transcript to affect any other transcript, and thus, we sum over all genes (with index $k$). Since most genes will not be regulatory, we use L1 regularization (LASSO) to shrink uninformative predictive coefficients to

zero. We also enforce a predicted rate of change of zero at time zero, reflecting the pre-induction steady-state assumption.

To arrive at this approach, we considered a suite of modeling strategies. We explored modeling dataset-level dynamics using a system of differential equations; however, such a model is both hard to fit and not robust to model mis-specification. Since a model of cellular regulation that *exclusively* includes transcriptional regulation is inherently incomplete, the parameters of such a model would be inappropriately contorted to compensate for in-expressible regulation. Regression models that express the measured abundance of a gene of interest based on measured abundances of candidate regulators do not suffer from such a problem. As many regression models can be posed, we explored a wide-space of model formulations defined by a set of hyperparameters (e.g., modeling in log- or linear-space, allowing for interaction terms, and adjusting regularization strength; see Appendix for complete details). To arrive at an optimal model formalism, we used cross-validation, whereby whole experiments were held-out and then predicted using all other experiments (encompassing 50 million regressions in aggregate).

Each regression model predicts the rate of change of a target gene based on other regulators (Fig 3A). These instantaneous estimates can be integrated to provide estimates of log$_2$ fold changes (Fig 3B). Grossly, the model explains 43% of the variability in log$_2$ fold changes (Appendix Fig S14). While the model does not account for all expression variability, both the dependent and independent variables are directly determined by experimental data. Accordingly, our inability to predict one gene's regulation does not affect modeling of other genes' regulation.

## Model coefficients predictive of edges in published networks

To assess the predictive power of the coefficients of this model on published datasets, we considered whether large regression effect sizes could predict interactions in two different genome-scale networks [a network of genetic interaction profiles, based on the growth of single and double mutants from Costanzo *et al* (Costanzo *et al*, 2016; Wang *et al*, 2004), and a probabilistic gene network that integrates multiple data types called YeastNet (Kim *et al*, 2014)] that may be enriched for true regulator–target relationships. The magnitudes of regression coefficients (from equation 1) are reasonable predictors of edges in both types of networks (AUC ~ 0.7), while early $t_{rise}$ times (from the CK model) do not have predictive power of genetic interactions in Costanzo *et al* (Fig 4A and B; Costanzo *et al*, 2016; Teixeira *et al*, 2018). A baseline LASSO regression model was included which fits each gene's expression as linear combinations of all genes' expression using a globally chosen λ. We find that our ODE-based model dominates a simpler regression baseline at every rank cutoff for the Costanzo dataset, outperforms the

baseline for YeastNet at stringent cutoffs, but underperforms the baseline model at more liberal cutoffs (Fig 4B).

Based on the ROC analysis, we next explored the strongest model coefficients that overlapped the genetic interaction profile similarity scores from Costanzo *et al*. This immediately revealed two interesting biological observations. First, Rpn4, the primary TF activator of yeast proteasome-encoding genes, is enriched for genetic interactions with proteasomal subunits (Costanzo *et al*, 2016; Wang *et al*, 2004; Appendix Fig S15). The straightforward interpretation of this result is that simultaneously removing the TF (Rpn4) and reducing the activity of individual subunits results in a negative interaction due to insufficient proteasome activity for supporting growth (Appendix Fig S15). Additionally, we found that many model-predicted targets of the S-phase-specific forkhead TF Hcm1 have interactions with *HCM1* (Costanzo *et al*, 2016). Loss of *HCM1* has been reported to increase the rate of chromosome loss (Pramila *et al*, 2006); analogous to the proteasome example, simultaneous loss of *HCM1* and Hcm1 target genes may further exacerbate this loss, resulting in negative genetic interactions. These results demonstrate that TFs and their targets can deviate from independence within published interaction networks because of direct functional regulatory interactions.

## Decomposing indirect regulation into sequential direct regulation

The parameters of equation 1 are regression coefficients that approximate $\partial y_i / \partial y_j$; in other words, they capture the potential of a gene $j$ to alter the expression of a gene $i$. Within the framework of our model, gene $j$ must also vary in expression to realize its regulatory potential.

To attribute changes to individual regulators, we consider their marginal contributions to overall timecourse changes. Using this approach, we look at each differentially expressed gene in a given experiment and attribute each regulatory response (e.g., rise, or fall) to one or more regulators based on regulators' marginal contributions to the response. Revisiting the Aft1 experiment, this marginal attribution analysis suggests that several regulators are operating in a cascade to regulate genes with different kinetics (Fig 3C). In line with binding data, Aft1 is predicted to be the primary regulator of early activated genes, while Aft1 is predicted to turn off genes (in part) through the activation of Hmx1. Each regulator–target relationship can be thought of as a directed edge in a graph, with the whole graph describing how the regulation is predicted to have unfolded across time during each induction experiment. Applying marginal attribution analysis to each experiment reveals that the induced TF is the primary driver of gene expression changes in most experiments (Appendix Fig S16); however, numerous other regulators are predicted mediators of indirect effects.

---

**Figure 3.   Predicted causal attribution of Aft1-driven transcriptional changes.**

A   Using a representative Aft1-responsive gene, *YHB1*, fold change differences between time points (solid gray dots) are compared to the LASSO regression model's fit (crosses). The model's predicted marginal contribution of three predicted regulators (Fet3, Hmx1, and Arn2) to the combinatorial control of *YHB1* is shown with bars that sum to the model's overall fit (crosses).

B   The *YHB1* fold change differences fit by regression can be converted to full timecourses to determine the marginal contributions of each regulator in driving a regulatory transition of interest (rises and falls).

C   Each differentially expressed gene in the Aft1 experiment is laid out based on its kinetics and colored according to the regulator predicted to be the strongest driver of differential expression. Model-derived fractional contributions of regulators to expression of *AQY2*, *NSR1*, *YHB1*, and *HEM25* are depicted as donut charts.

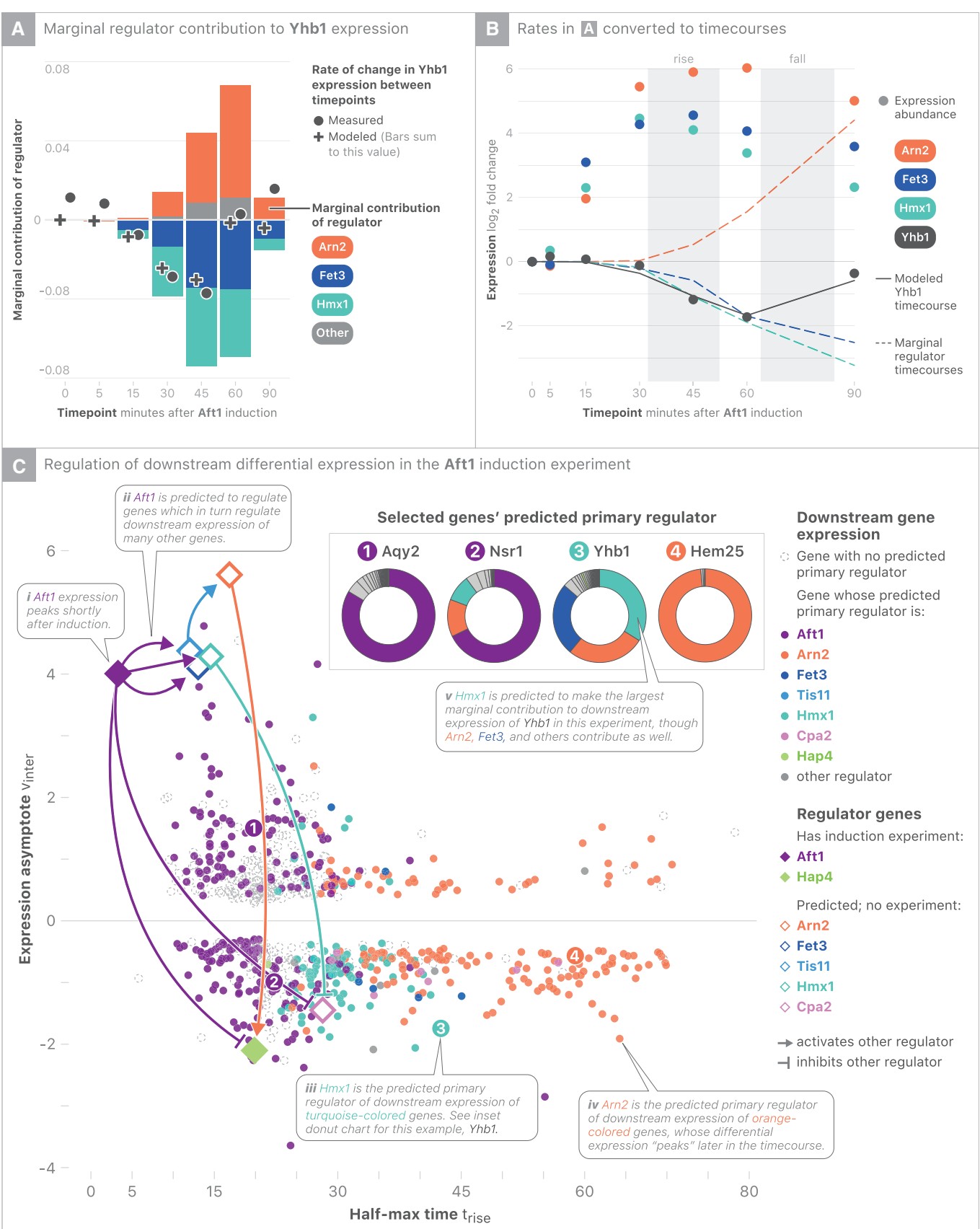

**Figure 3.**

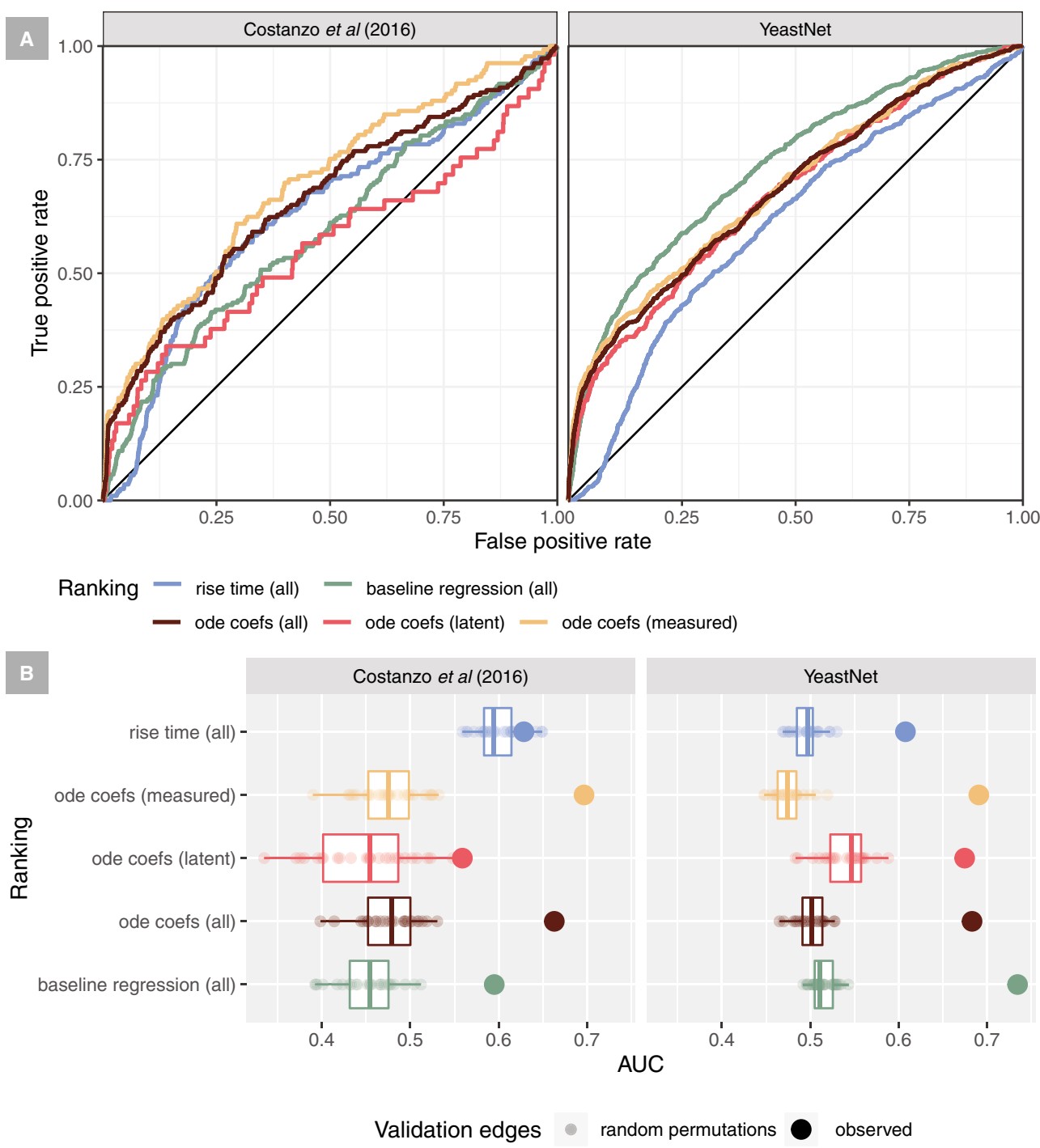

**Figure 4. Inferred regulator–target coefficient magnitudes are predictive of interactions in published datasets.**

The rankings of observed induction responses (ascending by $t_{rise}$) and predicted regulator relationships (descending by regression coefficient magnitude) were used to predict edges in two networks (Costanzo *et al*, 2016 and YeastNet v.3). All non-zero ODE regression coefficients were split into two classes: measured (predicted regulators with an induction experiment) and latent (predicted regulators without an induction experiment).

A    ROC curves are shown treating interactions, which overlap with the query edges (non-zero regression coefficients), as positives and all other entries as negatives.

B    The AUC of the ROC curves in (A) are shown as large dots, compared to 25 random permutations (which preserve the counts of each cause and effect but randomize their pairings) of the edge list which are shown as small dots and a boxplot of these null AUCs. Permutations with values systematically in excess of a value of 0.5, as seen when predicting genetic interactions with rise time, are likely due to hub effects where genes that tend to be predicted regulators of many targets also tend to have interactions. For boxplots, the central line is the median, hinges correspond to the first and third quartiles, and whiskers extend to the minimum and maximum values.

The synthesis of experiment-level graphs reveals a genome-scale causal expression network (Fig 5A) that links induced regulators to predicted intermediate regulators and downstream biological processes. This network reveals several regulatory hubs (including Hmx1, Stp4, and Fmp48), altered in numerous experiments, which are associated with consistent sets of downstream targets (Fig 5B).

## Regulatory potential of understudied genes

Our modeling results highlight many potential new regulators that we sought to confirm experimentally. These regulators include both hubs predicted to regulate targets across many experiments as well as mediators of interesting dynamic phenomena (such as impulses).

To validate putative regulators, a separate β-estradiol induction experiment was generated for each of ten candidate regulators (Dataset EV2).

Three out of ten of these induction experiments (Hmx1, Stp4, and Fmp48) showed strong changes in the putative regulators' targets ($P < 10^{-11}$; overlap of predicted and measured targets by chi-square and Fisher exact tests; Fig 6, Appendix Fig S17, Table EV3). One of the phosphate-related genes, Phm6, also appeared significant in the predicted and measured effects, but the experiment's expression dynamics were markedly similar to a mild stress response; thus, we did not interpret Phm6 as a likely transcriptional regulator. Model-driven discovery of three transcriptional regulators is notable, both because few genes are thought to be able

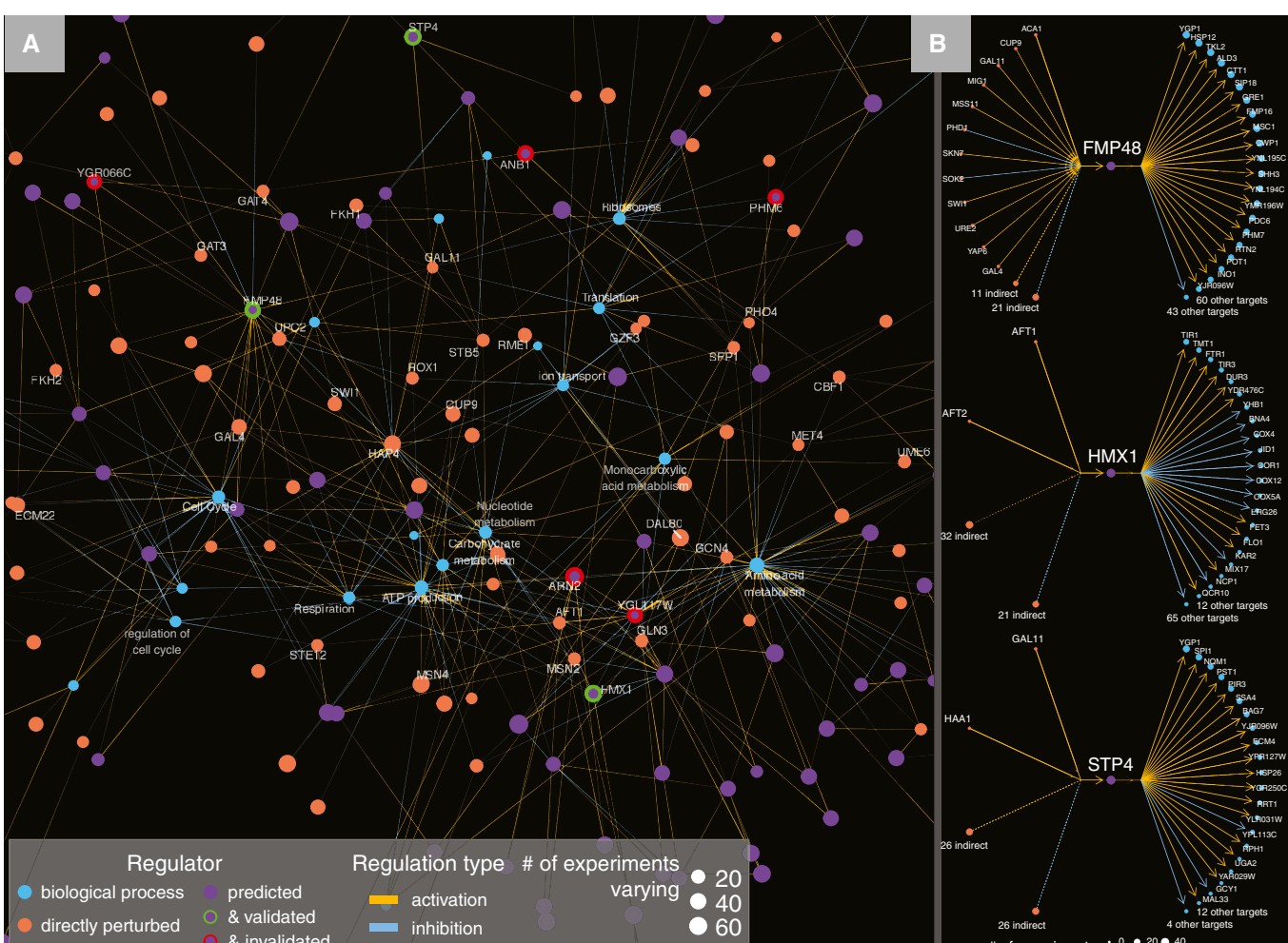

**Figure 5. Synthesis of predicted networks.**

Direct regulation between genes is defined based on causal attribution analysis, and indirect regulation of an induced gene is defined if a gene is differentially expressed regardless of whether attribution analysis indicated a direct regulatory relationship.

A   Edges between both genes with induction experiments and predicted regulators were formed based on regulatory interactions predicted from individual experiments (as shown in Fig 3C). For this visualization, major regulators selected as per Fig 3C are rooted to the induced TF regardless of whether they were directly or indirectly regulated by this gene. Predicted regulators are linked to GO categories based on having a significant overlap with their predicted targets. Similarly, genes with an induction experiment are linked to GO categories based on overlap of either direct or indirect targets with GO categories. Validated nodes (green) are genes where validation experiments confirmed a significant overlap between measured gene-regulator connections and model-predicted coefficients. Invalidated nodes (red) are genes where validation experiments failed to confirm model-predicted coefficients.

B   Local networks based on upstream direct/indirect regulators and downstream direct targets of three validated regulators.

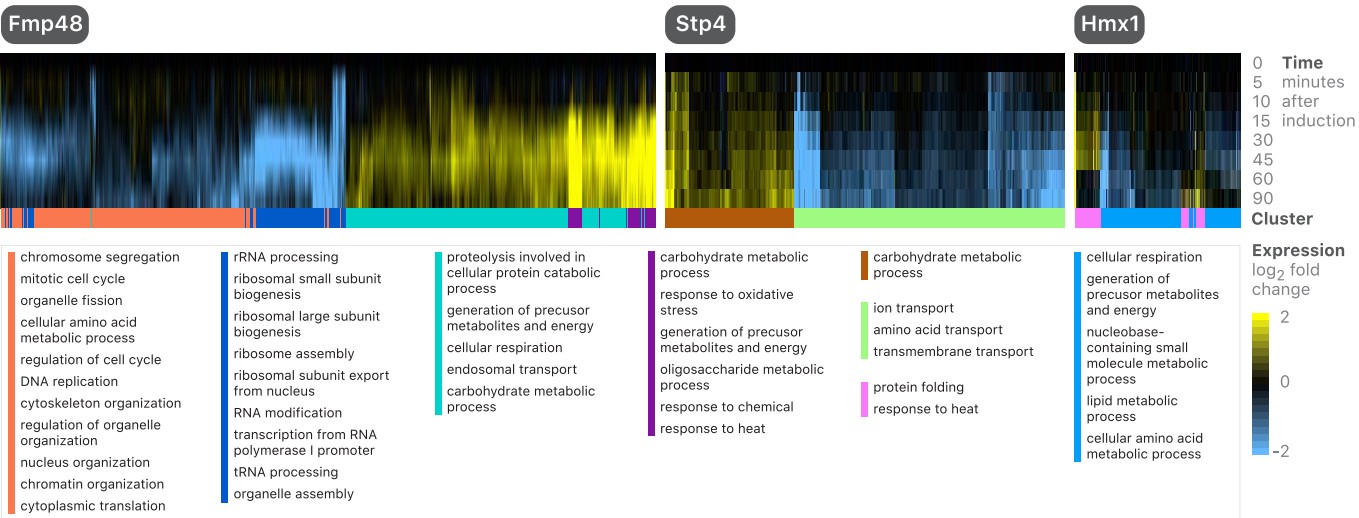

**Figure 6. Model-driven identification of transcriptional regulators.**
All genes passing hard-thresholding in each experiment are shown. K-means clustering was used to cluster responsive genes (K = 4 for Fmp48 and K = 2 for Stp4 and Hmx1), and GO slim gene-sets enriched in each cluster are shown.

to act as specific transcriptional regulators and because all confirmed regulators are poorly studied. In line with the integrative signals that we aimed to capture in this study (Fig 1E), the three confirmed regulators change in many experiments and act as hubs which connect diverse TFs to prominent regulatory processes. Each validated regulator temporally preceded its predicted targets. Additionally, variation in the regulator's $t_{rise}$ values were correlated with target $t_{rise}$ values (Appendix Fig S18A), and the regulator's $v_{inter}$ values were correlated with target $v_{inter}$ values (Appendix Fig S18B). Invalidated regulators, in contrast, either temporally coincided with their spurious targets or were active in a small number of experiments making them difficult to distinguish as true regulators among other correlated genes.

We find that as part of the iron utilization cascade, Aft1/Aft2 activate expression of *HMX1*. Previously, it was found that *hmx1Δ* cells accumulate heme (Protchenko & Philpott, 2003). Our results highlight a regulatory link between iron regulation, heme metabolism, and sterol synthesis in which Hmx1 induction results in the inhibition of *COX* genes, and genes involved in sterol biosynthesis (a process that requires heme), including *CYB5* (encodes cytochrome b5) and nearly every *ERG* gene (Smith *et al*, 1996; Appendix Fig S19A). Stp4 is annotated as a potential TF that contains a Krueppel-like domain (Cherry, 1998). Our data suggest that Stp4 is a *bona fide* TF resulting in activation/repression of many genes (Fig 6). Strongly repressed genes are enriched for the GNRCGGCY motif, consistent with previously published protein binding array data of Stp4 (Zhu *et al*, 2009; de Boer & Hughes, 2012); genes responsive to this motif are enriched for transmembrane transport (corrected *P*-value = 2.07 × 10$^{-8}$) (Appendix Fig S19B). Fmp48, named after "found in mitochondrial proteome", is a putative protein of unknown function with predicted kinase activity. Consistent with a previous study that revealed Fmp48 to be part of the TOR signaling network, we find corroborating evidence that Fmp48 is a transcriptional regulator that is as hub-like as the most highly connected individual TFs (Fig 6, Appendix Figs S19C and

S20; Breitkreutz *et al*, 2010). While the biological role of Fmp48 is understudied, our modeling approach and validation experiment confirmed it to be a regulator of gene expression.

## Discussion

To understand regulatory architecture, we require datasets that elicit diverse physiological regulatory responses, and possess sufficient information to disambiguate the drivers of each regulatory response. As demonstrated here, synthetic biology holds great promise for creating such datasets, and, when combined with new analytical tools, can be utilized to identify new regulators and GRNs.

Several results are worth highlighting. First, only a small number of genes annotated as being TF-bound, based on ChIP, respond in a typical experiment, and most responses are previously undescribed, including ~1,700 instances of ephemeral homeostatic impulses. Second, expression variation is associated with a modest number of transcriptional regulators with major effects, while many perturbations elicit minimal transcriptional responses. Thirty-eight TFs affected the expression of < 50 genes each (Appendix Fig S12A). Third, using kinetic information can help prioritize potential causal regulatory relationships. By integrating hundreds of timecourses with a dynamical systems model that explicitly includes time, we can make predictions of new regulatory interactions without the use of prior knowledge. Fourth, model coefficients can be used to predict interactions in published undirected networks. We expect that using IDEA to add weights and directionality to edges in existing networks will be powerful for understanding how cellular processes are dynamically coordinated in the cell.

Approaching genome-scale modeling of network regulation from dynamic data yielded insights about how to collect, process, and analyze such data. Because most genes do not respond in a typical induction experiment, we used hard-thresholding to remove the majority of values in our dataset, leaving ~ 100,000 gene-level

timecourses with coherent, biologically feasible patterns of variability. Having identified signal-containing timecourses, we were then able to model these signals using parametric fits and regularized regression. When fitting a genome-scale regression model, we made a number of assumptions about biological processes to include vs. those we should ignore based on our experimental design. In doing so, our model may fail to capture a number of important regulatory phenomena including complex combinatorial regulation, post-transcriptional regulation, post-translational regulation, localization, and regulation due to non-proteins (e.g., metabolites; Buchler *et al*, 2003; Bintu *et al*, 2005; Tan *et al*, 2011; Chong *et al*, 2015; Hackett *et al*, 2016). These phenomena could hamper analysis to the extent that regulators are absent, their concentrations are misrepresented, or their kinetics are temporally shifted. Because a transcriptional model is inherently incomplete, our modeling approach was structured to be robust to mis-specification by describing variables directly from data rather than creating latent variables. Our model builds relationships between genes with coherent regulatory relationships without being grossly biased by regulation that it cannot represent. An additional challenge we faced when fitting a model that allows for regulation by any gene is distilling a single regulator from a set of possibly highly correlated possibilities. Furthermore, some regulators not well represented by our dataset may be correlated with measured transcripts raising the possibility that predicted transcriptional regulators are false-positives if they are correlated with an unmeasured regulator. By utilizing > 200 experiments, we are in a regime where the correlation structure among genes begins to break down (Appendix Fig S13) and identifiability of an individual regulator among measured (and likely unmeasured putative regulators) becomes feasible. Future experiments should continue to erode this correlation structure, as most of our failed predictions (e.g., of the key drivers of Pho4 impulses) likely stem from this issue.

In this manuscript, we used synthetic perturbations and genome-scale time-series measurements to generate the IDEA dataset. Only one TF was perturbed at a time, resulting in a large but relatively sparse gene expression dataset. In the future, the use of combinatorial perturbations, as well as induction of non-TFs, will result in richer dynamic datasets. Indeed, as new timecourse datasets become available and are integrated with time-series analysis and prior knowledge, predictive models may require fewer experiments to build. Moreover, dataset generation and model evaluation naturally dovetail when using synthetic perturbations. Regulators can be easily tested with new induction experiments. When modeling predictions succeed, we confirm new biology; when they fail, the model gets better.

## Materials and Methods

### Growth conditions

For all experiments, cells were grown under continuous culturing conditions in 500-ml vessels as previously described with minor adjustments (Saldanha *et al*, 2004). Cultures were aerated with 6 l/min of humidified air at 30°C, maintained at 300 ml, and stirred with a magnetic impeller at 400 RPM. For the majority of experiments, cultures were maintained with minimal medium under phosphate limitation (20 mg/l). Nitrogen-limited cultures were maintained at 40 mg/l ammonium sulfate. Growth rates were maintained from 0.15 to 0.17/h. Batch growth in the chemostat vessels was initiated from a 1:60 dilution of a saturated overnight culture prior to turning on the chemostat pumps. Cells were grown to steady state, as determined by culture density, prior to the addition of 1 μM β-estradiol to the culture and subsequent sampling. Chemostat experiments were performed with either the Infors Sixfors or the Multifors systems.

### Strain construction

Parent strains were engineered to constitutively express an artificial transcription factor that is inducible with estradiol. Parent strains used for gene expression analysis contain either the GEV or the $Z_3EV$ transcription factor. A synthetic promoter fused to selectable marker (typically KanMX) of ~ 2 kilobases (kb) in length was PCR-amplified and introduced into parent strains with homologous recombination using a standard lithium acetate transformation procedure. For the vast majority of strains, clones containing the synthetic promoter were selected for on rich medium [YPD (1% yeast extract, 2% bacto-peptone, and 2% dextrose) containing G418 (200–300 μg/ml)]. Primers were designed using custom software in R such that the cassette was introduced directly between a target gene's first methionine residue and its native promoter to prevent the removal of any genomic DNA. Synthetic promoters were inserted into the genome without removing native DNA for two reasons. First, we believed that removing at TF's native promoter could disrupt expression of a divergently transcribed gene. Second, binding sites in *Saccharomyces cerevisiae* need to be within a few hundred base pairs of an ORF to be functional (Dobi & Winston, 2007). Therefore, in our case, displacement of the native promoter by ~ 2 kb is likely to remove its regulatory potential of the TF-encoding gene.

### Extraction, labeling, and hybridization of RNA

Crude RNA was extracted using an acid–phenol procedure. RNA was then purified using either the QIAGEN RNeasy kit or the RNAClean Ampure XP beads. 200 ng of cleaned RNA was used as an input to generate dye-labeled cRNA using the Agilent Quick-Amp Labeling Kit. Labeled cRNA was cleaned using RNAClean Ampure XP beads or RNeasy. Reference RNA was extracted from DBY12001, a laboratory wild-type strain, grown to steady state in a phosphate-limited chemostat at D ~ 0.18/h. RNA from references and samples was labeled with Cy3-CTP and Cy5-CTP, respectively. Labeled RNA was hybridized to Agilent 8 × 15k microarrays, which were then washed, scanned, and processed using the Agilent Feature Extraction software with default settings and loess dye bias correction.

### Datasets

We performed a number of signal processing steps to our microarray datasets. In increasing order of processing, we refer to these as the "raw" dataset, "cleaned" dataset, "noise-model thresholded" dataset, and the "shrunken" dataset.

Each microarray has some number of spots (usually 2) for each gene. For each spot, we computed ratio = max(red, C)/max(green, C), where C = 2 in arbitrary units. This minimum value application

only affects 0.3% of spots. Typical genes have red and green channel measurements above 200. These (red, green, ratio) values for each spot serve as the "raw" data.

For each microarray, we aggregated the data across individual spots. Specifically, for each gene at a given time point within an experiment we aggregated the spot values and measured the minimum value, maximum value, median value, and standard deviation of the values. In the usual case of two spots, the median value is equivalent to mean and standard deviation is equivalent to (max−min)/sqrt(2).

At this stage, we corrected the most extreme outlier observations. First, for a given sample, we examined the case where, for a given gene, the ratio of the spot values was larger than four. Since the spots values do not agree, we interpolated the value with the geometric mean across bracketing time points (or neighboring time point in the case of first or last time point). The second class of outliers is where the median ratio moves by a factor of at least four between two time points and then by a factor of four in the opposite direction for the next time point. In these cases, we again replaced the central point with the geometric mean of the bracketing points. These corrections apply to less than 0.2% of the data.

In processing gene expression microarrays, crosstalk between red and green channels can occur. When the red channel fluorescence is much larger than the green channel fluorescence, there can be leakage of signal, and the green channel measurement is affected. To identify instances of this occurring, we first computed the green channel ratio relative to the time zero measurement. We then measured the 30% quantile of this value within each timecourse (for time points after $t = 0$), as well as the 30% quantile of the log-ratio. We then flagged timecourses where the green ratio quantile exceeded a factor of eight, and the log-ratio quantile exceeds a twofold change. These represent cases where the green channel is increasing a great deal when it should be constant. For these rare cases, we repaired the red-to-green ratios by duplicating the time zero green channel across the full timecourse. This affects about two-dozen TF-gene timecourses (out of more than a million).

The GEV and ZEV systems both have characteristic gene expression signatures, including a mild stress response. Previously, it has been shown that singular value decomposition (SVD) is one way to remove such signals (McIsaac et al, 2012). Here, we used a slightly different approach. First, we computed the median time series for each gene in each class of experiments (GEV and ZEV). We then subtracted this median time series, leaving a normalized log-ratio. Because a given gene is not directly or indirectly affected by a transcription factor in most experiments, the median is an accurate reflection of any background time-dependent behavior. We refer to the dataset where outliers are removed and the GEV/ZEV signal is removed as the "cleaned" dataset.

Further details on data processing and modeling can be found in the Appendix.

### De novo motif discovery

To construct data-driven binding motifs for each induced TF and suggest other TFs that may be operating in each induction experiment, we sought to identify cis-regulatory motifs which are enriched in the promoters of regulated genes and, where possible, attribute a known regulator to those motifs. To identify enriched motifs, we utilized the regular-expression-based software, DREME (Bailey, 2011), to identify short (8-mers or shorter) motifs that are enriched in a set of primary sequences relative to a set of control sequences. The sequences we focused on for all motif analyses were the promoters of yeast genes (defined as 500 base pairs upstream of each gene; downloaded from Ensembl on 2018-01-03). To identify regulatory motifs, for each experiment, sequences in the promoters of regulated genes (sigmoid or impulse responses) were compared to non-differentially expressed control sequences. Similarly, motifs associated with impulse-like behavior were identified by identifying motifs enriched in the promoters of genes with impulse kinetics, using the promoters of genes with sigmoidal kinetics as control sequences. To attribute identities to each motifs, probability weight matrices (PWMs) based on binding data were downloaded from Yeastract and Cis-BP and then matched to each DREME motif using TOMTOM (Gupta et al, 2007; Weirauch et al, 2014; Teixeira et al, 2018).

To investigate whether genes containing an enriched motif exhibit stereotypical kinetics, we investigated whether variation in regulatory kinetics across genes could be predicted based on promoter composition. To carry-out such comparisons, the promoters of regulated genes were matched to each identified motif based on enrichment of high-scoring PWM k-mers (Pr(sequence|PWM)) in the promoters of regulated compared to control promoters (using the same primary vs. control comparisons as for DREME). Briefly, this was done by ordering sequences in descending order of PWM score match and then using a rolling mean estimate of k-mer frequency to estimate the PWM score cutoff where enrichment in the primary sequences decreases to a heuristic 1.5-fold cutoff. Using this approach, each gene responding in a given experiment was summarized based on how many times each motif was detected and what its strongest PWM match was. To determine whether any motif was associated with variation in a kinetic property ($v_{inter}$, $t_{rise}$, and rate for regulated genes, with $v_{final}$ and $t_{fall}$ added for impulse genes), each kinetic coefficient was regressed on a motif-by-motif basis using ordinary least squares (OLS) on three summaries of each motif's presence. These three predictors were as follows: the top enriched PWM match for the gene, a binary variable indicating whether one or more motifs were present, and counts of how many PWM matches were found. OLS t-statistic P-values were separately FDR-controlled for each type of motif summary ("best match", "motif present", "# of matches") and model type (regulation or impulse) (Storey & Tibshirani, 2003).

### Dynamical system modeling overview

We pursued a linear regression approach to modeling equation 1. First, we constructed an estimator of the time derivative of the gene expression response, which is treated as the dependent variable. We then fit a linear model to extract the coefficients of the dynamical system. This works because the time derivatives of the gene expression levels are modeled as linear functions of gene expression levels (possibly with quadratic terms as well). We note that this does not actually correspond to a full solution of the dynamical system, but requires point-wise consistency with the dynamical system description. Selection of regularization levels with cross-validation yielded a model for the transcriptional effects of gene expression levels. This model was interrogated to identify which regulators were most

important for predicting observed expression changes in each time-course. Derivations implementing this modeling approach are presented in the Appendix sections: Linear Regression, BIC regularization, Hyperparameter Search, and Cross-Validation.

### Marginal attribution analysis

The whole-cell regression model readily provides two important summaries of regulation. First, the estimated coefficients of the regression model, $\alpha$ and $\beta$, capture regulatory potential (i.e., $\alpha = \partial y/\partial x$). Second, the regression model combines regulatory potential with variation in regulators through observation-level fitted values ($\alpha X$) to summarize how regulation unfolded within an experiment (where $x_{ij}$ is the abundance of a gene $i$ at time $j$ for a given experiment). To interpret regulation, we want to be able to identify the major variable regulators underlying regulatory phenomena of interest such as the impulse-like dynamics of Aft1 or variable timing of expression induction or repression. Since realized regulation is a property of an experiment, the fitted model ($\alpha X$) informs whether the model collectively predicts regulatory expression changes of interest and marginal interpretation of components of this model ($\alpha_i x_{i.}$) can be used to attribute regulation to specific regulators.

To attribute regulation, we first identify instances of regulation that are reasonably predicted by the whole-cell model. Each instance of realized regulation is a change in expression occurring over a period of time. These transitions in the data can be readily understood within the framework of the previously discussed parametric models since these models indicate which regulatory phenomena to track and the saturation of the sigmoidal curves implies the period of time over which regulation is unfolding: $t\{\text{sat} = x\} = t_{\text{coef}} + \log(x/(1 - x))/\alpha$.

Here, $t\{\text{sat} = x\}$ is the time at which the sigmoid is $x$ saturated (i.e., 90% of the response having occurred equates to $x = 0.9$). $t_{\text{coef}}$ is the half-max time coefficient of the transition ($t_{\text{rise}}$ or $t_{\text{fall}}$ for a given phenomena). With this convention, the end-points of each rise and fall phenomena, $t_{\text{start}}$ and $t_{\text{end}}$, were defined as the time when the rise or fall sigmoid was 5–95% saturated.

To determine when the whole-cell model accounts for an appreciable fraction of observed regulatory changes, model-predicted fold changes over each regulatory interval $f^{\text{model}} = \log_2(y[t_{\text{end}}]) - \log_2(y[t_{\text{start}}])$ were compared with the observed change in that gene's expression over the regulatory interval $f^{\text{observed}} = \log_2(x[t_{\text{end}}]) - \log_2(x[t_{\text{start}}])$. Since $\{t_{\text{start}}, t_{\text{end}}\}$ will generally occur between time points, linear interpolation of both $\log_2(y)$ and $\log_2(x)$ with the two closest time points was used to infer these intermediate expression states. For 47,802 responses (rises or falls), the model had some predictive value based on the following cutoff:

$$\min([f^{\text{model}}, f^{\text{observed}}])/\max([f^{\text{model}}, f^{\text{observed}}]) > 0.2$$

Marginal attribution analysis was used to dissect total model fits into the marginal contributions of each regulator. Here, the marginal attribution of each regulator to the response was defined as:

$$\psi_{ijkz} = \frac{\left| f_{ijkz}^{\text{model}} \right|}{\sum \left| f_{ikz}^{\text{model}} \right|}$$

Here, $\psi_{ijkz}$ is the model's predicted proportional control of a regulator $j$ to a gene $i$ in experiment $k$ for the $z^{\text{th}}$ response (rise or fall)

occurring in that timecourse. $\psi$ values (filtered to $\psi > 0.2$) were used to interpret the major regulator(s) contributing to observed responses in individual experiments and in the meta-graph of cross-experiment regulation.

## Data and software availability

Datasets and computer code used in this study are publicly available.

- Microarray data, Gene Expression: https://www.ncbi.nlm.nih.gov/geo/query/acc.cgi?acc = GSE142864; http://idea.research.calicolabs.com.
- Computer software, Bayesian Chechik & Koller Model: https://github.com/calico/impulse.
- Computer software, Dynamical System Model: https://github.com/google-research/google-research/tree/master/yeast_transcription_network.

**Expanded View** for this article is available online.

### Acknowledgements
We thank Ali Bashir, David Botstein, Rochelle Buffenstein, Chiraj Dalal, Michelle Dimon, Dan Gottschling, and John Platt for comments and critical feedback on the manuscript.

### Author contributions
SRH, EAB, MC, and RSM performed analysis and wrote the paper. MB and MF performed analysis. BJW, GK, DGH, and RSM performed experiments. EAB, MC, and MB developed the dynamical system model. SRH wrote the *impulse* R package. AB developed the interactive website. RSM conceived and oversaw the study.

### Conflict of interest
SRH, DGH, BJW, GK, AB, and RSM are employees of Calico Life Sciences LLC. EAB, MC, MF, and MB are employees of Google LLC. The authors declare that they have no conflict of interest and no competing financial interests from this work.

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
