## [Review Process File · Molecular Systems Biology]

Learning causal networks using inducible transcription factors and transcriptome-wide time series

Scott McIsaac, Sean Hackett, Edward Baltz, Marc Coram, Bernd Wranik, Griffin Kim, Adam Baker, Minjie Fan, Marc Berndl, and David Hendrickson

Review timeline:

Submission date:	13 th August 2019
Editorial Decision:	16 th August 2019
Re-submission received:	26 th October 2019
Editorial Decision:	11 th December 2019
Revision received:	10 th January 2020
Editorial Decision:	6 th February 2020
Revision received:	13 th February 2020
Accepted:	19 th February 2020

Editor: Jingyi Hou

Transaction Report:

1st Editorial Decision

16th August 2019

Thank you for having submitted a manuscript entitled "Learning regulators from IDEA, a set of hundreds of dynamic transcriptome-wide induction experiments" for consideration for publication in Molecular Systems Biology.

Your paper has now been seen by Editors of the Journal, and we have decided to return it to you without sending it for extensive peer review.

In this study you present IDEA, a gene expression time-course dataset monitoring the response to the individual induction of hundreds of transcription factors (TFs) in yeast. You further present a computational framework for inferring direct regulatory connections among genes, based on integration of all time-course data in IDEA and computational modeling, without requiring prior information. We appreciate that you identify novel transcriptional regulators and that you experimentally validate three previously unknown transcriptional hubs that are predicted in your model. However, we think that in absence of a global assessment of the obtained results and a direct comparison of IDEA to existing alternative methods, the validity of the identified regulatory connections and hubs seems somewhat tentative, and the potential of IDEA and the presented computational approach for deriving novel biological insights that would not be possible to obtain using existing workflows, remains to be demonstrated. Overall, we are not convinced that study provides the kind of broadly relevant resource and the kind of decisive methodological advance with demonstrated potential to generate novel biological insight that would be required for publication at Molecular Systems Biology.

Authors' re-submission

26th October 2019

Based on your comments, we've gone back and benchmarked our whole-cell modeling predictions in a few ways. We've now shown that not only can we predict new regulators, but we can also predict the presence of genetic interactions within two different published networks (Yeastnet and Costanzo). This ability of using a model of transcription to predict gene-gene interactions (with AUC ~ 0.7), is quite unprecedented -- furthermore, it highlights new biology that we now discuss in great detail. Additionally, we've benchmarked our dataset against ChIP and published gene expression data of mutants, which highlights that dynamic perturbation data is more functionally coherent than other data types with far fewer data sets.

Given the results of these new analyses, we were hoping you would consider a resubmission.

2nd Editorial Decision

11th December 2019

Thank you again for submitting your work to Molecular Systems Biology. We have now heard back from two of the three reviewers who agreed to evaluate your manuscript. Unfortunately, after a series of reminders we did not manage to obtain a report from reviewer #3. In the interest of time, and since the recommendations of referee #1 and #2 are quite similar, I prefer to make a decision now rather than further delaying the process. As you will see below, the reviewers acknowledge that the presented approach and dataset seem potentially useful for the field. They raise however a series of concerns, which we would ask you to address in a major revision.

I think that the recommendations of the reviewers are rather clear so there is no need to repeat the points listed below. All issues raised by the reviewers need to be convincingly addressed. As you may already know, our editorial policy allows in principle a single round of major revision, so it is essential to provide responses to the reviewers' comments that are as complete as possible. Please feel free to contact me in case you would like to discuss in further detail any of the issues raised by the reviewers.

REFEREE REPORTS

Reviewer #1:

Summary

In this paper, Hackett et al have taken a previously-published system for small-molecule transcriptional induction system in *Saccharomyces cerevisiae* and applied it to more than 200 transcriptional controllers (transcription factors, chromatin modifiers, etc). The authors constructed strains, performed genome-wide gene expression profiling using DNA microarrays over several short-term timepoints to identify the genes that change in expression as a result of overexpression, and these expression changes were used to solve dynamic models that correspond to regulator-gene connections. Parameters of these solved models were then used as evidence for edges in a gene regulatory network.

General Comments

This is an important and well-executed study. The overall experimental design is excellent and represents a major improvement over comparable methods that typically use gene deletion alleles in steady-state to identify TF targets. The resulting dataset is of very high quality and of considerable immediate value to researchers in the field. The core analysis of this data is rigorous and well done. I expect that this work will be of great interest for developing computational regulatory network inference tools in the short term, and this work will stand as a major advance in our understanding of the transcriptional regulatory network in yeast. I have some comments on the data presentation and analysis which, if addressed, will enhance this work. I would also like to specifically commend the interactive presentation of this data set.

Major

1. The authors state that the target TF is over-expressed 53-fold relative to $t=0$. Does this mean that there is some leaky expression in the uninduced state? How does this compare to the fold induction of natural inducible systems like the GAL or NCR systems. It would be useful to have some idea of how overexpressed each TF is relative to wildtype levels.
2. The authors don't provide a rationale for why cells were grown in chemostats. Although I am sure that they would argue that this means the cells are in "steady-state" I think a stronger argument for the use of chemostats would help. Moreover, does the condition in which the experiment is performed have an effect on the observed induction? For example, in Figure S9 suggests that many genes that lie below the diagonal are metabolic-related TFs. And in Figure S4 it appears that PHO4 has the largest number of impulse responses - is this because the cells are grown in phosphate-limited chemostats. Ideally, the authors should provide experiments that illustrate the extent to which the method is sensitive to the conditions by over-expressing the same TF in multiple conditions.
3. I was surprised that the strain construction did not replace the endogenous regulatory sequence, but inserted the inducible system regulatory sequence downstream of the endogenous sequence. Do the authors have evidence that this doesn't result in any problems e.g. can GAL4 still be induced effectively in rich glucose media?
4. Supplemental figure 11 shows a large number of experiments which have few or no changes in gene expression. In the discussion it states that ~40 TFs had no effect. It would be valuable to explore if those induced genes that give no response have anything in common (e.g. requiring some stoichiometry with other proteins, some highly-controlled posttranslational processing like proteolytic cleavage, etc)
5. The validation experiment is confusing in presentation, and the information content of figure 6 seems to be quite low for a main text figure. The authors' model has predicted network edges associated with 10 potential regulators; it is necessary to include confusion matrixes for all of the validation experiments and a reasonable model performance metric (chi-square does not seem appropriate). In addition I would like to see the differential expression heatmaps (as in Figure 5) for all 10 confirmation experiments (as supplemental figures). Induction experiments which result in no gene expression changes are trivially explained as regulators that do not function correctly after induction and overexpression; experiments where there are gene expression changes that are not predicted by the initial model are much more interesting.
6. Functional coherence doesn't seem like a valid way to benchmark gene regulatory networks. The underlying rationale is shaky, especially when using GO slim terms (is an enrichment of 'response to chemical' meaningful?). I also have a general concern about using differences between p-values as statistical evidence outside the context of hypothesis testing. The ROC curves associated with S14 are much more compelling (I am surprised they are not main-text figures).
7. The supplemental methods package related to the model-fitting is excellent (I found the answer to every margin note that I made about the modeling approach while reading the results section). However, the main-text methods section lacks details of a number of analyses that are presented in the manuscript; I believe that as written it would be very difficult to replicate some of the supporting analyses in this work. The authors should provide more detail on these methods.
8. In Figures S3 and S4 there are other very obvious horizontal and vertical banding patterns that are difficult to interpret. It would help to have these better annotated. The figure legend simply points to one of these bands.
9. In all of the heatmaps that the authors present some genes go up and some genes go down. Is this expected - are these TFs known to have repressive activity? Are these all due to secondary biological effects? Can the authors rule out that data normalization doesn't play a role in this effect?
10. The website is very nice. However, I was surprised that the dataset is not presented as a global

gene regulatory network "at the scale of the entire genome" (which is the phrase the authors use to motivate their study).

Minor

1. In the introduction the authors state that "by not utilizing prior knowledge, we minimize bias against re-learning known biology". This seems counterintuitive to me - incorporating prior knowledge builds on years of work and effort, and "re-learning" known biology is a means of validating the approach. Is there a way to rephrase this statement?
2. It would be useful to the reader to have an intuitive explanation of the Chechik and Koller kinetic model.
3. The authors use 'timecourse' in different contexts; the intro has timecourses which are experiments ("We generated over two-hundred TF induction timecourses"), and results seems to switch between timecourses which are experiments and timecourses which are gene measurements in an experiment ("The signals from these 100,036 timecourses were retained"). The terminology used needs to be clearly defined and preferably not reused. I found this to be very confusing when first reading the manuscript.
4. It would be helpful to include a brief introduction to Aft1 and the motivation for selecting that gene as the focus.
5. The results section refers to Rpn4 and cites supplemental figure 14; Rpn4 is not in that figure and it's not clear where the Rpn4 claims originate.
6. Figure 4 has annotations for validated & invalidated regulatory nodes without any further explanation. What makes a node valid or invalid? It's also not clear how regulatory pathways (ontologies?) have been integrated into this network.
7. The yeasttract citation isn't compiled and the eQTL study (ref 17) is incomplete.

Reviewer #2:

Review of "Learning causal regulatory networks with inducible promoter alleles and massively parallelized time series measurements".

Overall I thought this paper was great, and describes a great dataset and an interesting analysis. It hurt a bit to see how disconnected the paper was from prior work, and a major comment is that more work needs to be done to properly frame this in the context of the mature field it naturally fits into. I am quite positive about this paper and only have so many critical comments due to my keen interest in the topic.

A main flaw is that the network inference was not sufficiently well described, I want to hear more about alternatives, why you chose this model, and have a proper formulation of the network inference in the methods section.

p.2. Constructing these GRNs typically requires extensive prior knowledge [3,4].

More direct citations are available.

p. 2. How can we develop the regulatory clarity of GRNs at the scale of the entire genome?

What is Clarity?

What does "Hyper-ChIP-able" mean ? ... much too informal and inexact. Say less, but be specific.

"Additionally, we find that 79% of genes reported as being directly bound by a TF do not exhibit a significant expression response in the corresponding TF's induction experiment"

"In IDEA, realized regulation is directly measured and there is stronger agreement between early

induction events in IDEA with TF-DNA interactions as assessed with transposon calling cards than by ChIP "

No, many TFs need to be post transcriptionally or post translationally modified to be active. This could be due to interactions. TFs do not act alone. Also where are you getting these numbers ?

Binding is not a great proxy for "regulates" I agree there, mostly suffering from false positives. But over expression dynamics will still have a huge false negative rate due to post-X mods not being lined up.

Figure 3c ... I can't figure it out. This could be part my problem and part the figure's problem, but perhaps take a peek at this one.

Functional coherence is used as a proxy for regulates. This seems like a very bad idea.

Compare known motifs (Cis-BP and FiMo) to your de novo analysis.

"we fit a Bayesian version of the Chechik & Koller kinetic model to each 5 timecourse [23]"
... please explain this more thoroughly.

On p. 10 you, while discussing the use of other networks (a genetic interaction network and a functional association network) state : "To our surprise, the magnitudes of regression coefficients are reasonable predictors of edges in both types of networks (AUC ~ 0.7), while early t rise times do not have predictive power of genetic interactions (Figure S14) [22,28]."

1st Revision - authors' response

10th January 2020

Reviewer #1:

Summary

In this paper, Hackett et al have taken a previously-published system for small-molecule transcriptional induction system in *Saccharomyces cerevisiae* and applied it to more than 200 transcriptional controllers (transcription factors, chromatin modifiers, etc). The authors constructed strains, performed genome-wide gene expression profiling using DNA microarrays over several short-term timepoints to identify the genes that change in expression as a result of overexpression, and these expression changes were used to solve dynamic models that correspond to regulator-gene connections. Parameters of these solved models were then used as evidence for edges in a gene regulatory network.

General Comments

This is an important and well-executed study. The overall experimental design is excellent and represents a major improvement over comparable methods that typically use gene deletion alleles in steady-state to identify TF targets. The resulting dataset is of very high quality and of considerable immediate value to researchers in the field. The core analysis of this data is rigorous and well done. I expect that this work will be of great interest for developing computational regulatory network inference tools in the short term, and this work will stand as a major advance in our understanding of the transcriptional regulatory network in

yeast. I have some comments on the data presentation and analysis which, if addressed, will enhance this work. I would also like to specifically commend the interactive presentation of this data set.

Major

1. The authors state that the target TF is over-expressed 53-fold relative to t=0. Does this mean that there is some leaky expression in the uninduced state? How does this compare to the fold induction of natural inducible systems like the GAL or NCR systems. It would be useful to have some idea of how overexpressed each TF is relative to wildtype levels.

We thank the reviewer for making these points and agree they are important to address. To determine how a TF's expression compares between engineered and WT strains, we've added a new informatic analysis. Specifically, we obtained the red and green channel values for each TF from the t = 0 and t = 90 min. timepoints from its respective induction experiment. Additionally, for each TF, we calculated its red/green ratio across all experiments to create a distribution for each TF. Since the vast majority of these ratios come from strains where the TF is under its native control, the median of these distributions should provide an estimate of how much the TF level varies between or experimental strains and our universal reference WT strain. For each TF, the median ratio was ~1, indicating that we can directly compare the normalized red and green channels from the t = 0 minute sample to estimate the level of leakiness and 90 minute sample to estimate the fold-change above WT. These data are shown in a newly added **Appendix Figure S2**. We estimate that the TF expression in 86% of our synthetic-promoter driven TF strains is lower than that in WT at t = 0 min. At t = 90 min, the median TF is 28.4-fold above WT TF levels. We've added these data to the manuscript near the beginning of the Results section. The legend of the new figure reads, "*Appendix Figure S2: Estimating leakiness and inducible of synthetic promoter-driven TF alleles. For each TF strain, the red (sample) and green (reference) microarray values were obtained from the t = 0 min. and t = 90 min. samples. The red/green ratio provides an estimate of leakiness for the t = 0 min. histogram (blue) in which 86% of synthetic promoter-driven TFs have expression less than WT TF levels. At t = 90 min., the red/green ratio provides an estimate of induction above WT TF levels (red histogram). The median level of TF induction over WT TF levels is 28.4.*"

GAL overexpression can result in a 500-1000-fold increase in expression. Exploring a recently published dataset that looked at how patterns of gene expression change in response to a large pulse of a preferred nitrogen source, we estimate the repressibility of NCR genes to be at least 100-fold. We've added this result to the manuscript as a point of comparison for the interested reader. We now write: "*Induction of a target TF is detectable in <5 minutes and reaches saturation within ~10 minutes following β -estradiol addition at a median level 53-fold higher than at t = 0 min (Appendix Figure S1B), which, in terms of magnitude, is less than the ~500-1000-fold inducibility of Gal4-driven promoters in the presence of galactose and similar to the ~50-100-fold repressibility of Nitrogen Catabolite Repression (NCR) genes in response to an increase in nitrogen availability [34-*

36J.” We have also added three new references related to the GAL and NCR results, which are listed below.

Johnston M. A model fungal gene regulatory mechanism: the GAL genes of *Saccharomyces cerevisiae*. *Microbiol Rev.* 1987;51: 458–476.

Airoldi EM, Miller D, Athanasiadou R, Brandt N, Abdul-Rahman F, Neymotin B, et al. Steady-state and dynamic gene expression programs in *Saccharomyces cerevisiae* in response to variation in environmental nitrogen. *Mol Biol Cell.* 2016;27: 1383–1396.

Biggar SR, Crabtree GR. Cell signaling can direct either binary or graded transcriptional responses. *The EMBO Journal.* 2001. pp. 3167–3176.
doi:10.1093/emboj/20.12.3167

2. The authors don't provide a rationale for why cells were grown in chemostats. Although I am sure that they would argue that this means the cells are in "steady-state" I think a stronger argument for the use of chemostats would help. Moreover, does the condition in which the experiment is performed have an effect on the observed induction? For example, in Figure S9 suggests that many genes that lie below the diagonal are metabolic-related TFs. And in Figure S4 it appears that PHO4 has the largest number of impulse responses - is this because the cells are grown in phosphate-limited chemostats. Ideally, the authors should provide experiments that illustrate the extent to which the method is sensitive to the conditions by over-expressing the same TF in multiple conditions.

We thank the reviewer for these important points. We address each of them in full below.

Although I am sure that they would argue that this means the cells are in "steady-state" I think a stronger argument for the use of chemostats would help.

We completely agree with the reviewer’s comment. In the original submission, we only wrote that cells are in steady state and provided no further context for why that actually matters. We’ve modified the beginning of the results section to emphasize the relevance of the steady-state condition. We’ve added, *“We chose chemostats, in part, because the steady-state condition of chemostat cultures is a particularly useful feature for mathematical modeling. Under steady-state conditions, the levels of molecules and activities of processes are not changing at a culture-wide level. Therefore, following TF induction in a steady-state culture, immediate dynamic changes result from the TF induction itself. The ability to choose a single growth-limiting nutrient also makes the chemostat ideal for exploring how input-output relationships between TFs and target genes vary under different nutritional conditions [29].”*

Moreover, does the condition in which the experiment is performed have an effect on the observed induction?

The reviewer makes a very good point in asking if the growth condition affects the expression response. The answer to this question is “yes”. We previously performed induction experiments of the methionine TFs (Met4, Met28, Met31, Met32, and Cbf1) under methionine limitation. We found that Cbf1 would switch from an activator (under methionine limitation) to a repressor (under phosphate limitation where cells have excess extracellular methionine). Additionally, there are other targets in the genome where Cbf1 is an activator independent of methionine limitation. For methionine metabolic genes, the regulatory connections exist in both conditions -- but the sign of those edges (positive versus negative) is different. We’ve added the following text to the introduction: *“This study also revealed that Cbf1 could act as an activator or a repressor, depending on which promoter it targets [29]. For certain methionine metabolic genes, Cbf1 can act as an activator of target genes when yeast are limited for methionine, but can switch to being a repressor of those same genes when yeast are limited for phosphate and have excess extracellular methionine [29]. These results highlight the ability of TF induction to reveal condition-dependent regulatory connections, and that a TF can act as both a positive and negative regulator of gene expression depending on local DNA context and environmental conditions.”*

Ideally, the authors should provide experiments that illustrate the extent to which the method is sensitive to the conditions by over-expressing the same TF in multiple conditions.

This is a very interesting point. We previously published a study with the methionine TFs, but didn’t discuss in great detail in our original submission - this was an oversight on our part. We have expanded the introduction to explain those results. TF activity can certainly depend on condition, and our induction approach has been used to reveal such dependencies. In IDEA, we also have induction experiments for multiple nitrogen TFs performed under both phosphate and nitrogen limitation. They are in the dataset, but were also not discussed specifically in our original submission. We now include some discussion of those data, and highlight them for readers particularly interested in nitrogen metabolism. In short, the nitrogen TFs looked remarkably similar in both conditions (unlike the Cbf1 example). In the case of Gln3, we interpreted the similar responses to mean that it wasn’t the limitation that mattered (nitrogen versus phosphate), but rather the source of nitrogen itself, since we used ammonium sulfate as a nitrogen source in both experiments. Had we used a poorer nitrogen source (like proline), the expression responses to Gln3 activation may have been quite different. In the Results section we had add, *“Finally, IDEA also contains several TFs induced under multiple conditions. Gln3, Dal80, and Gzf3 were induced under phosphate and nitrogen limitation (with ammonium sulfate used as the sole nitrogen source in both cases). For each TF, the resulting expression patterns are strikingly similar in the two tested environments (Appendix Figure S3), suggesting that the activity of nitrogen-related TFs may depend more on the quality of the nitrogen source (proline vs. ammonium sulfate, for example), rather than the choice of growth-limiting nutrient.”*

3. I was surprised that the strain construction did not replace the endogenous regulatory sequence, but inserted the inducible system regulatory sequence

downstream of the endogenous sequence. Do the authors have evidence that this doesn't result in any problems e.g. can GAL4 still be induced effectively in rich glucose media?

We thank the reviewer for this comment, and agree that explaining/justifying insertion versus replacement is important. The promoter constructs we used are ~2kb in length and include a drug selectable marker linked a synthetic promoter that is placed upstream of a TF. In a strain that has been transformed with our construct, the TF's native promoter is 2kb away from the TF open reading frame (ORF). Since upstream activating sequences are normally only a few hundred basepairs from ORFs they regulated in *Saccharomyces cerevisiae*, the conventional wisdom is that the native promoter, in our engineered strain, is too far from the TF ORF to affect its expression - this is quite different than the situation in human cells, for example. The lack of long-range activation of genes in yeast by upstream activating sequences (UASs) has been quantified by Fred Winston's group with several UAS-TATA constructs (Dobi and Winston, 2007).

The lack of activation over long distances is one reason for using the insertion strategy. But additionally, by not removing native DNA, if there is a gene that shares a promoter with the TF and is divergently expressed (such as in the case of *GAL1* and *GAL10* sharing a common promoter region), its upstream regulatory region should remain unperturbed using the insertion strategy. We've added a reference to the Dobi and Winston paper, as well as the following text to the materials and methods: *"Synthetic promoters were inserted into the genome without removing native DNA for two reasons. First, we believed that removing at TF's native promoter could disrupt expression of a divergently transcribed gene. Second, binding sites in S. cerevisiae need to be within a few hundred base pairs of an ORF to be functional [63]. Therefore, in our case, displacement of the native promoter by ~2 kb is likely to remove its regulatory potential of the TF-encoding gene."*

The last part of the reviewer's comment refers to Gal4 -- specifically, *can Gal4 be induced and function in glucose-rich medium?* To address this concern, we have included a Gal4 experiment in IDEA, and concluded that the answer is "yes". We found that Gal4 can indeed turn on Gal4 target regions rapidly in our growth conditions (phosphate limitation with 2% glucose). On the IDEA website, if one goes to the "Induction heatmaps" section and types in "GAL4" they can see the genes that respond most strongly to Gal4 induction. *GAL7* and *GAL1*, for example, respond within 15 minutes to Gal4 induction with β -estradiol in our experimental conditions.

4. Supplemental figure 11 shows a large number of experiments which have few or no changes in gene expression. In the discussion it states that ~40 TFs had no effect. It would be valuable to explore if those induced genes that give no response have anything in common (e.g. requiring some stoichiometry with other proteins, some highly-controlled posttranslational processing like proteolytic cleavage, etc)

We thank the reviewer for this comment. We've modified the text slightly to read *“Thirty-eight TFs affected the expression of <50 genes each (Appendix Figure S12A).”* Furthermore, we've modified the figure legend to highlight which TFs we are referring to for the interested reader to explore more deeply. We do not see an obvious connection to explain the sparseness of the expression responses following induction of these particular TFs - they regulate diverse biological processes, and a few are annotated to act in complexes.

5. The validation experiment is confusing in presentation, and the information content of figure 6 seems to be quite low for a main text figure. The authors' model has predicted network edges associated with 10 potential regulators; it is necessary to include confusion matrices for all of the validation experiments and a reasonable model performance metric (chi-square does not seem appropriate). In addition I would like to see the differential expression heatmaps (as in Figure 5) for all 10 confirmation experiments (as supplemental figures). Induction experiments which result in no gene expression changes are trivially explained as regulators that do not function correctly after induction and overexpression; experiments where there are gene expression changes that are not predicted by the initial model are much more interesting.

We thank the reviewer for these important points. We address each of them in the revised manuscript. We've included a new supplementary figure (**Appendix Figure S17**), which shows heatmaps for all 10 validation experiments. The model edge predictions are also incorporated into this heatmap (shown in blue and red). We also completely agree with the reviewer on the importance of including confusion matrices between predictions and results. We now include these confusion matrices as **Table 5**. We also now include an additional statistical assessment of significance using Fisher's exact test. Finally, we've moved Figure 6 to the supplement, based on the reviewer's suggestion.

6. Functional coherence doesn't seem like a valid way to benchmark gene regulatory networks. The underlying rationale is shaky, especially when using GO slim terms (is an enrichment of 'response to chemical' meaningful?). I also have a general concern about using differences between p-values as statistical evidence outside the context of hypothesis testing. The ROC curves associated with S14 are much more compelling (I am surprised they are not main-text figures).

Based on comments from both reviewers, we've removed the analysis on “functional coherence” from the revised manuscript. We've moved Figure S14 from the original submission to the main text as the reviewer suggested (it is now **Figure 5**).

7. The supplemental methods package related to the model-fitting is excellent (I found the answer to every margin note that I made about the modeling approach while reading the results section). However, the main-text methods section lacks details of a number of analyses that are presented in the manuscript; I believe that as written it would be very difficult to replicate

some of the supporting analyses in this work. The authors should provide more detail on these methods.

We thank the reviewer for pointing this out, and are glad that the model-fitting methods “ticked all the boxes”. We tried to be extremely fastidious about describing the computational methods in detail. Some of the methods were placed in the main text, while others were placed in the Appendix.

We have made several substantive changes that we hope will address the reviewers concerns. First, in our newly added “Data Accessibility” section, we now include links to the code on Github for both the Chechik & Koller curve fitting model (<http://https://github.com/calico/impulse>) as well as the dynamical systems modeling that is used to solve Equation 1 in the main text (https://github.com/google-research/google-research/tree/master/yeast_transcription_network).

Second, we’ve moved more of the methods from the Appendix to the main text. Specifically, the summary of the dynamical systems modeling implementation has been moved. We’ve also added a sentence that points the interested reader to the Appendix for a complete and very technical accounting of the entire dynamical systems modeling approach. That, combined with the code provided above, will allow those who are interested to perform this type of analysis. Additionally, we’ve moved the entire methods section describing “marginal attribution analysis” used in Figure 3 to the main text as well.

8. In Figures S3 and S4 there are other very obvious horizontal and vertical banding patterns that are difficult to interpret. It would help to have these better annotated. The figure legend simply points to one of these bands.

We thank the reviewer for pointing this out. We’ve added more annotations to the figures as well as descriptions within the legends. We now clearly state (in the legend of Figure S3) that the vertical banding patterns are actually TFs that are hubs (they affect the expression of many genes when induced). The TF experiments are sorted alphabetically, and we highlight a subset of them including GAT3, GAT4, GCN4, MSN2, MSN4, SFP1, and UME6. The horizontal banding patterns are due to weak time-dependent signals that accompany most experiments (e.g., ESR-related). We highlight two of the most striking bands. The top horizontal band is due to the high variation of expression observed in ORFs that have no standard ORF ID beyond their chromosome and position (many of these are dubious ORFs). We’ve now clearly labeled those in both figures. The second large clear band is ribosomal genes, which we also now clearly annotate in both figures as well. It’s also worth pointing that color palette saturates at $\log_2 = -1$ and $\log_2 = 1$. We chose this dynamic range to accentuate some of the features in the raw data. We think that the labels the reviewer requested are a significant improvement, and will be useful to readers in interpreting these heatmaps.

9. In all of the heatmaps that the authors present some genes go up and some genes go down. Is this expected - are these TFs known to have repressive activity? Are these all due to secondary biological effects? Can the authors rule out that data normalization doesn't play a role in this effect?

TFs can act as direct activators and repressors of gene expression (like Cbf1). The full extent to which dual function (activation and repression) of TFs exist is unknown, but in cases where we see a clear binding motif associated with the induced regulator the motif tends to be overrepresented as either activating or inhibiting. This contrasts with some of the non-TF regulators that we identified from modeling and later validated. This is because Fmp48 is a putative kinase and activates some targets while inhibiting others.

We think that the vast majority of transcriptional changes elicited in this dataset are due to indirect responses to TF induction, and are interpreted as transcriptional cascades. As an example, Gln3, a well-studied strong transcriptional activator involved in nitrogen metabolism, represses many genes through indirect means. We see additional cases of indirect regulation, particularly those manifesting as “perfect adaptation”, which often involve acute changes in genes involved in ribosomes or anabolism. We think that some of these effects are mediated through global factors that we do not directly measure such as amino acid pools.

We have attempted to remove signals that are driven by gene- and array-level noise as well as removing stress-dependent patterns. Therefore, we think there are few examples of large dynamic changes which could be attributed to mis-normalization. The retention of signal is partially demonstrated by the replicate heatmaps of different TFs (notably Gln3), which have clear concordance when strong signal occurs. We describe all of the signal processing steps in great detail, and make the data available at all levels of processing for others to explore and develop alternative methods for interrogating signal of interest.

10. The website is very nice. However, I was surprised that the dataset is not presented as a global gene regulatory network "at the scale of the entire genome" (which is the phrase the authors use to motivate their study).

We thank the reviewer for making this point, and we’ve corrected this oversight in the revision. We have added an interactive version of our network from **Figure 4** to the website (<https://idea.research.calicolabs.com/network>). The Cytoscape file itself can be downloaded at <https://idea.research.calicolabs.com/data>.

We expect that future work from us and others will continue to refine how we explore and visualize these data to drive new discoveries. We want to point out that we also did some work to integrate IDEA with the Costanzo genetic interaction network, and this can be found under the “TF effects” panel on the website. We’ve made movies of how regulator-gene connections form over time on this network. While some TFs regulate clear “galaxies” within the genetic interaction “universe”, others do not. Understanding how to better integrate these kinds of data, and especially how different processes are coordinated to give rise to a healthy cellular state, is something we are keen to explore in the future, and believe others will too.

1. In the introduction the authors state that "by not utilizing prior knowledge, we minimize bias against re-learning known biology". This seems counterintuitive to me - incorporating prior knowledge builds on years of

work and effort, and "re-learning" known biology is a means of validating the approach. Is there a way to rephrase this statement?

We agree with the reviewer that this sentence was poorly phrased, and have removed the offending text. We now say: *“Our approach implicitly dissects indirect regulation into a series of direct regulatory relationships. Predicted intermediate regulators span canonical transcriptional regulators and genes of unknown function.”*

2. It would be useful to the reader to have an intuitive explanation of the Chechik and Koller kinetic model.

We thank the reviewer for this comment. The second reviewer also suggested that we add a more complete explanation of this model in the main text, and we agree that this would be useful for readers. We have left the detailed equations in the supplement, and have added a general explanation of the Chechik & Koller (CK) model to the main text. Specifically, we have added the text shown below in purple:

“As is the case in the Aft1 experiment, timecourses with significant signal across IDEA typically exhibit either a sigmoidal or impulse-like response (double sigmoidal); thus, we fit a Bayesian version of the Chechik & Koller (CK) kinetic model to each timecourse [42,43] (see materials and methods for more details on curve fitting; code for implementing CK fits can be found at <https://github.com/calico/impulse>). The CK model characterizes a timecourse as a double sigmoid but can be reduced to a simpler sigmoid that has fewer parameters. Specifically, the original CK kinetic model contains six parameters, which we reduced to five parameters because the initial amplitude for all timecourses is zero due to normalization. The impulse (double sigmoid) response is ideal for capturing two-transition behavior in biological timecourses. One sigmoid characterizes the onset response and a second sigmoid characterizes the offset response [42]. Parametric fits enable direct comparisons of timecourses by revealing kinetic parameters. Our Bayesian implementation ensures that these parameters are interpretable by penalizing unrealistic and impossible parameterizations (e.g., step-function responses or changes which precede β -estradiol introduction). Since the impulse and single sigmoid models are nested (i.e., the simpler model contains all of the terms within the more complex model), we can - for a given timecourse - use a likelihood ratio test to determine if extra parameters improve the fit sufficiently to justify the more complex model.

Sigmoidal responses are summarized with a half-max time constant (t_{rise}), an asymptotic expression level (v_{inter}), and a slope parameter (β). Impulses include two additional parameters: t_{fall} , which describes the time when the response returns halfway to its final level, and v_{final} , the asymptotic expression level of the impulse (Figure 2B) [43].”

3. The authors use 'timecourse' in different contexts; the intro has timecourses which are experiments ("We generated over two-hundred TF induction timecourses"), and results seems to switch between timecourses which are experiments and timecourses which are gene measurements in an experiment ("The signals from these 100,036 timecourses were retained"). The terminology used needs to be clearly defined and preferably not reused. I found this to be very confusing when first reading the manuscript.

We thank the reviewer for pointing this out, and completely agree that this needs to be clearer. We have gone through the text and now use “experiment” to refer to the set of all gene expression responses that follow the induction of a single TF. Additionally, we now use “timecourse” to specifically refer to a particular genes’ expression response in a single experiment. At the beginning of the Results section we have added, *“In this manuscript, an experiment refers to all of the gene expression responses that follow from induction of single TF. A timecourse refers to the kinetic response of a single gene within a single experiment.”*

4. It would be helpful to include a brief introduction to Aft1 and the motivation for selecting that gene as the focus.

We thank the reviewer for making this suggestion, and we’ve added the following text to manuscript where introduce Aft1: *“Aft1 was originally identified as an activator of genes that uptake iron into the cell [38]. Aft1 responds to defects in iron-sulfur cluster biogenesis [39], and its activity is negatively regulated by Met4, the primary activator of methionine biosynthetic genes [39,40]. We highlight Aft1, in part, because we observe a range of expression responses following its activation.”*

It’s also worth noting that we also decided to focus on Aft1 due to our modeling predictions, which indicated that Hmx1, (part of the Aft1 regulon) was itself a regulator of gene expression.

5. The results section refers to Rpn4 and cites supplemental figure 14; Rpn4 is not in that figure and it's not clear where the Rpn4 claims originate.

We thank the reviewer making this comment. We sought to clarify this result in the revision by adding both text and an additional figure. The observation that Rpn4, which activates proteasomal subunits, genetically interacts with many of its targets, came from the ROC analysis shown in what is now **Figure 5** (Figure S14 from our original submission). It has long been known that Rpn4 up-regulates proteasomal subunits (and thereby proteasomal activity). In IDEA, Rpn4 robustly activates these genes as well. What hasn’t been appreciated is that Rpn4 genetically interacts with many of its targets. The ROC curves (comparing the magnitudes of coefficients from our transcriptional model with two published networks [Yeasnet and Costanzo]) revealed this connection. We’ve added a new supplemental figure (**Appendix Figure S15**) to show *rpn4Δ*’s strongest interactions from Costanzo (using the interactive tool available at <http://thecellmap.org>). We also modified the text to explain how we made these observations. Specifically, we added, *“Based on the ROC analysis, we next explored the strongest model coefficients that overlapped the genetic interaction profiles from Costanzo et al. This immediately revealed two interesting biological observations.”*

6. Figure 4 has annotations for validated & invalidated regulatory nodes without any further explanation. What makes a node valid or invalid? It's also not clear how regulatory pathways (ontologies?) have been integrated into this network.

We thank the reviewer for noting these issues as being unclear in our initial submission. In Figure 4, validated nodes are the genes we induced in our validation experiments and saw a significant overlap between responding genes and gene-regulator connections within the model. In equation, those connections are represented as “alpha” values. We have modified the legend of the figure to clearly state what “valid” and “invalid” mean in this particular context. Specifically, we now write, “*Validated nodes (green) are genes where validation experiments confirmed a significant overlap between measured gene-regulator connections and model-predicted coefficients. Invalidated nodes (red) are genes where validation experiments failed to confirm model-predicted coefficients.*” The GO categories are meant to simplify the network and make it more “human readable”. If a regulator is connected to hundreds of targets, we pick the GO term with highest significance to visualize rather than the hundreds of targets individually. We’ve modified the legend to include, “*Predicted regulators are linked to GO categories based on having a significant overlap with their predicted targets.*”

7. The yeasttract citation isn't compiled and the eQTL study (ref 17) is incomplete.

We thank the reviewer for pointing out these errors. We’ve fixed them both. The eQTL reference is shown below.

Lutz S, Brion C, Kliebhan M, Albert FW. DNA variants affecting the expression of numerous genes in trans have diverse mechanisms of action and evolutionary histories. *PLoS Genet.* 2019;15: e1008375.

Reviewer #2:

Review of "Learning causal regulatory networks with inducible promoter alleles and massively parallelized time series measurements".

Overall I thought this paper was great, and describes a great dataset and an interesting analysis. It hurt a bit to see how disconnected the paper was from prior work, and a major comment is that more work needs to be done to properly frame this in the context of the mature field it naturally fits into. I am quite positive about this paper and only have so many critical comments due to my keen interest in the topic.

We greatly appreciate the reviewer's positive view of our work, and as detailed below, we've expanded the text (including the introduction) to better frame this paper in the context of previous work.

A main flaw is that the network inference was not sufficiently well described, I want to hear more about alternatives, why you chose this model, and have a proper formulation of the network inference in the methods section.

We thank the reviewer for this comment, and we have sought to improve our formulation of the network inference approach in the main text (both in the Results section and the Methods section). The full model derivation, we believe, is beyond the scope of the main text, so we relegate it to the following four sections in the Appendix: Linear Regression, BIC regularization, Hyperparameter Search, and Cross-Validation. But we also now clearly highlight these sections for the reader in the main text, which was missing from our original submission. Additionally, in the revision, we have included a link to code in a public repository for implementing the network inference approach. To the main text Results section, we've added the following text to explain the approach and discuss alternatives, as suggested by the reviewer:

"To arrive at this approach, we considered a suite of modeling strategies. We explored modeling dataset-level dynamics using a system of differential equations; however, such a model is both hard to fit and not robust to model mis-specification. Since a model of cellular regulation that exclusively includes transcriptional regulation is inherently incomplete, the parameters of such a model would be inappropriately contorted to compensate for in-expressible regulation. Regression models that express the measured abundance of a gene of interest based on measured abundances of candidate regulators do not suffer from such a problem. As many regression models can be posed, we explored a wide-space of model formulations defined by a set of hyperparameters (e.g., modeling in log- or linear-space, allowing for interaction terms, and adjusting regularization strength) (see Appendix for complete details). To arrive at an optimal model formalism we used cross-validation, whereby whole experiments were held-out and then predicted using all other experiments (encompassing 50 million regressions in aggregate)."

Furthermore, in the main text Methods section, we now include a section call "Dynamical systems modeling overview", which reads: *"We pursued a linear*

regression approach to modeling Equation 1. First, we constructed an estimator of the time derivative of the gene expression response, which is treated as the dependent variable. We then fit a linear model to extract the coefficients of the dynamical system. This works because the time derivatives of the gene expression levels are modeled as linear functions of gene expression levels (possibly with quadratic terms as well). We note that this does not actually correspond to a full solution of the dynamical system, but requires point-wise consistency with the dynamical system description. Selection of regularization levels with cross-validation yielded a model for the transcriptional effects of gene expression levels. This model was interrogated to identify which regulators were most important for predicting observed expression changes in each timecourse. Derivations implementing this modeling approach are presented in the Appendix sections: Linear Regression, BIC regularization, Hyperparameter Search, and Cross-Validation.”

p.2. Constructing these GRNs typically requires extensive prior knowledge [3,4].

More direct citations are available.

We thank the reviewer for this comment, and we completely agree with it. In our original attempt at brevity, we missed a number of references as well an opportunity to properly frame our work in the context of previous work on GRNs. We have added many references (in total, the revised manuscript has 66 references, compared to 44 in the original submission), and now give a more detailed description of early work on the inference of GRNs and networks motifs. We’ve also cited more recent work on using mutant libraries and CRISPR tools to identify GRNs in multiple organisms.

We’ve added the text in purple to the Introduction: *“The direct and indirect molecular interactions that achieve a particular cellular state can be described as regulatory edges that collectively form Gene Regulatory Networks (GRNs) [3,4]. As genome-scale datasets started to become available over 20 years ago, work by Alon and colleagues established that certain GRN topologies are enriched in biological systems [4]. Understanding the functional properties of such “network motifs” became the subject of intense experimental and theoretical investigation [4–14]. Combined with genomic tools and extensive prior knowledge, it became possible to identify network motifs/GRNs associated with core cellular processes, with early work in yeast focusing on cell cycle control and the DNA damage response [15,16]. The widespread development and adoption of genome-scale technologies, including the creation of mutant libraries and the power of CRISPR-Cas systems, has further enabled GRN discovery in living organisms, from plants [17], to yeast [18], to humans [19].”*

We have also added a reference and description of a beautiful recent paper (Solis *et al*) that used a combination of genomic tools and time series analysis using the “anchor away” method to identify a core GRN within the Hsf1/proteostasis network. Specifically, we’ve added the following text: *“Integration of multiple*

'omic technologies combined with time series measurements can help identify direct functional interactions to elucidate GRNs, as was done in a recent study that combined RNA-seq, NET-seq, and ChIP-seq to identify a core regulon for Hsf1 in yeast [28].'

p. 2. How can we develop the regulatory clarity of GRNs at the scale of the entire genome? What is Clarity?

We have removed this sentence, and agree with the reviewer that our original phrasing was confusing. We have modified the offending sentence to say, *"How are genomic approaches commonly applied to identify GRNs?"*

What does "Hyper-ChIP-able" mean ? ... much to informal and inexact. Say less, but be specific.

We thank the reviewer for pointing this out - our phrasing was too colloquial, and not adequately explained. We removed the word "hyper-ChIPable", and modified the text to read, *"Target genes with similar ChIP profiles can exhibit opposite expression responses [26], and highly expressed portions of the genome can exhibit strong ChIP signal even amongst unrelated proteins [27]. Interpreting the biological importance of such peaks must be done with sufficient controls to distinguish whether signals are truly biological versus technical in origin, but the challenge remains that ChIP-based approaches alone provide no assessment of TF functionality."*

"Additionally, we find that 79% of genes reported as being directly bound by a TF do not exhibit a significant expression response in the corresponding TF's induction experiment". "In IDEA, realized regulation is directly measured and there is stronger agreement between early induction events in IDEA with TF-DNA interactions as assessed with transposon calling cards than by ChIP ". No, many TFs need to be post transcriptionally or post translationally modified to be active. This could be due to interactions. TFs do not act alone. Also where are you getting these numbers ?

We thank the reviewer for this question/points. We tackle them, one at a time, below.

"In IDEA, realized regulation is directly measured and there is stronger agreement between early induction events in IDEA with TF-DNA interactions as assessed with transposon calling cards than by ChIP ". No, many TFs need to be post transcriptionally or post translationally modified to be active. This could be due to interactions. TFs do not act alone.

We've removed the above statement in the revision, and clarified the text to read, *"Additionally, we find that 79% of genes reported as being directly bound by a TF based on published ChIP measurements do not exhibit a significant expression response in the corresponding TF's induction experiment (Appendix Figure S10)*

[41]. *The low recall of reported transcriptional regulation underscores the value of dynamic data. Realized regulation may be impacted by chromatin accessibility and the regulatory context of the extracellular environment, which can result in different post-translational modifications of TFs [29,44–46].*”

Also where are you getting these numbers?

We’ve added a citation to Yeabstract in the main text, and clarified that we are calculating the percentage of previously identified binding sites that result in measurable changes in expression of target genes in response to TF induction. The vast majority of these sites don’t result in realized regulation in our dataset. We don’t want to over-interpret this result, because there are biological and technical reasons a difference could be observed that are beyond the scope of the current study. Therefore, we just write “...79% of genes reported as being directly bound by a TF based on published ChIP measurements do not exhibit a significant expression response in the corresponding TF’s induction experiment (Appendix Figure S10) [41].”

Binding is not a great proxy for "regulates" I agree there, mostly suffering from false positives. But over expression dynamics will still have a huge false negative rate due to post-X mods not being lined up.

We thank the reviewer for making these important points. We’ve modified the introduction substantially to address this, and explicitly discuss Cbf1, which some of us previously showed can switch between activating or repressing methionine genes depending on environmental conditions. An interesting future analysis could be to compare binding to induction experiments and look for TFs that have the largest disagreements across the genome between binding and expression effects. This kind of “computational screen” could identify TFs that likely require modification to be active under various conditions. We’ve also added a supplemental figure for Gln3, which is an activator regardless of whether or not the cells are limited for nitrogen or phosphate. As mentioned above, we’ve also added some discussion of *Solis et al.* that combined RNA-seq, NET-seq, ChIP-seq, and time series analysis (using the “anchor away” method) to identify a core GRN for Hsf1/proteostasis. This multi-omic approach was focused on one TF, but also provides a useful framework for exploring functional versus non-functional TF binding.

Figure 3c ... I can't figure it out. This could be part my problem and part the figure's problem, but perhaps take a peek at this one.

We thank the reviewer for pointing this out. This figure does indeed contain quite a bit of information. To help the reader navigate this figure, we now include five “speech bubbles” which explain what each part of the figure, and provides the reader an order in which to view different parts of the figure (i, ii, iii, iv, and v). For (i), we now specify “*Aft1* expression peaks shortly after induction”, and point to the purple dot that represents *Aft1*. *Aft1* is strongly activated so its v_{inter} value from the CK model is much greater than 0. Then, in (ii), we now write “*Aft1* is predicted to regulate genes which in turn regulate downstream expression of many other

genes". This speech bubble points directly at 3 genes (Fet3, Tis11, and Hmx1), which are predicted intermediate regulators. In the third (iii) speech bubble, we write "*Hmx1 is the predicted primary regulator of downstream expression of turquoise-colored genes. See inset donut chart for this example, Yhb1.*" For (iv), we write, "*Arn2 is the predicted primary regulator of downstream expression of orange-colored genes, whose differential expression "peaks" later in the timecourse.*" For (v), we write, "*Hmx1 is predicted to make the largest marginal contribution to downstream expression of Yhb1 in this experiment, though Arn2, Fet3, and others contribute as well.*"

We believe that these will help make the figure much easier to understand.

Functional coherence is used as a proxy for regulates. This seems like a very bad idea.

Based on comments from both reviewers, we've removed this analysis from the revised manuscript.

Compare known motifs (Cis-BP and FiMo) to your de novo analysis.

In our original submission, we included a table that included the motifs we discovered with DREME, including matches to known motifs. The known motifs (PWMs) were downloaded from the Yeastract database and matched to the motifs we identified with DREME using the TOMTOM software package (which, like FiMo, is part of the MEME suite of software tools). *Based on the reviewer's comment, we've repeated this same analysis using motifs from Cis-BP.* This results are nearly identical, and we've included them as part of **Table 3** in our revision. We've also added a reference to Cis-BP to the main text (Weirauch *et al*, 2014).

"we fit a Bayesian version of the Chechik & Koller kinetic model to each timecourse [23]"... please explain this more thoroughly.

We thank the reviewer for this comment. Indeed, a better explanation of the Chechik & Koller model was requested by both reviewers. The text we added to fully explain the model is shown above in response to the first reviewer's comments. Additionally, we have added a link to all of the code for performing the sigmoid and impulse fits (<http://github.com/calico/impulse>).

On p. 10 you, while discussing the use of other networks (a genetic interaction network and a functional association network) state : "To our surprise, the magnitudes of regression coefficients are reasonable predictors of edges in both types of networks (AUC ~ 0.7), while early trise times do not have predictive power of genetic interactions (Figure S14) [22,28]."

It looks like this comment was cut off, but we've made some minor changes to this section aimed at clarifying our findings. We now write, "*The magnitudes of regression coefficients (from Equation 1) are reasonable predictors of edges in both types of networks (AUC ~ 0.7), while early trise times (from the CK model) do*

not have predictive power of genetic interactions in Costanzo et al. (Figure 4) [41,48]. A baseline LASSO regression model was included which fits each gene's expression as linear combinations of all genes' expression using a globally chosen λ ."

3rd Editorial Decision

6th February 2020

Thank you for submitting your revised manuscript to Molecular Systems Biology. We have now heard back from the two reviewers who agreed to evaluate your manuscript. As you will see the reviewers are now overall supportive and I am pleased to inform you that your manuscript will be accepted in principle pending the following essential amendments:

1. Reviewer #1 has expressed concerns about the use of the word "timecourse" in the text, please address this properly.
2. Please address reviewer #2's concern by improving the introduction and/or discussion sections in light of previously published work.

REFEREE REPORTS

Reviewer #1:

The revised manuscript has addressed all of my comments. I have one remaining concern that should be addressed in the text; I don't believe that any additional experimental or analytic work is necessary.

The authors have continued to use 'timecourse' to refer both to whole genome expression changes through an experiment, and to single gene expression changes through an experiment. In the results section, they have defined these terms to be "In this manuscript, an experiment refers to all of the gene expression responses that follow from induction of single TF. A timecourse refers to the kinetic response of a single gene within a single experiment." Figure 2A clearly refers to a timecourse as whole genome expression changes, and figure 2B clearly refers to a timecourse as a single gene expression change through an experiment. Many of the textual uses of 'timecourse' are ambiguous in context and could be read either way (changing the interpretation of several key parts of this work). Based on the definition provided, I read "The Aft1 timecourse is an illustrative example of the value of induction data for revealing intricate regulatory phenomena" as the change of Aft1 expression, but it seems more likely to be the experiment where Aft1 is induced. It is absolutely essential that the terminology used be consistent throughout this work, including figure legends and captions (like Appendix Figure S3, 5, & 6).

Reviewer #2:

The authors have responded to most of my comments. They still fall short on connecting this to prior works, but have made minor improvement on that front. Overall I was positive prior and remain positive given the improvements in the paper. I am in favor of publishing this work.

2nd Revision – authors' response

13th February 2020

Reviewer #1:

The revised manuscript has addressed all of my comments. I have one remaining concern that should be addressed in the text; I don't believe that any additional experimental or analytic work is necessary.

The authors have continued to use 'timecourse' to refer both to whole genome expression changes through an experiment, and to single gene expression changes through an experiment. In the results section, they have defined these terms to be "In this manuscript, an experiment refers to all of the gene expression responses that follow from induction of single TF. A timecourse refers to the kinetic response of a single gene within a single experiment." Figure 2A clearly refers to a timecourse as whole genome expression changes, and figure 2B clearly refers to a timecourse as a single gene expression change through an experiment. Many of the textual uses of 'timecourse' are ambiguous in context and could be read either way (changing the interpretation of several key parts of this work). Based on the definition provided, I read "The Aft1 timecourse is an illustrative example of the value of induction data for revealing intricate regulatory phenomena" as the change of Aft1 expression, but it seems more likely to be the experiment where Aft1 is induced. It is absolutely essential that the terminology used be consistent throughout this work, including figure legends and captions (like Appendix Figure S3, 5, & 6).

We thank the reviewer for pointing this out. We've carefully gone through the manuscript and made sure that the language is consistent throughout the main text and appendix. We've edited Figures S3, S5, and S6 to include the word "experiment" and not "timecourse".

Reviewer #2:

The authors have responded to most of my comments. They still fall short on connecting this to prior works, but have made minor improvement on that front. Overall I was positive prior and remain positive given the improvements in the paper. I am in favor of publishing this work.

We are pleased that we addressed the majority of the reviewer's comments in the first revision. We had added a number of references to papers on GRN inference and discovery in the previous revision. In this revision, we've significantly expanded the introduction to spell out work from the DREAM project on network inference. Specifically, we added, "Finally, there is a growing literature of computational methods for reconstructing GRNs from high-throughput data [29–35]. The Dialogue on Reverse Engineering Assessment and Methods (DREAM) project, which is organized around annual challenges, provides a framework to benchmark network inference methods [29]. Network inference performance can depend on implementation as well as the network structure itself [31]. In the DREAM5 challenge, no single inference method performed optimally across multiple datasets. Integrating predictions across all participating teams (35 inference methods in total) to generate "community networks" had the most robust performance [31]." Additionally, we added a reference to Chua *et al.* in the Introduction: "A seminal paper from Chua *et al.* revealed that overexpression of a single TF, followed by transcriptome profiling at a single time point, can reveal functional regulator-gene connections that are absent when profiling TF deletion mutants [36]. Following that work, we combined TF activation with dynamic transcriptome profiling to dissect the incompletely understood regulatory connectivity of the yeast sulfur regulon [37]."

Accepted

19th February 2020

Thank you again for sending us your revised manuscript. We are now satisfied with the modifications made and I am pleased to inform you that your paper has been accepted for publication.

Corresponding Author Name: Scott Mclsaac

Manuscript Number: MSB-19-9174RR